# NEK9 regulates primary cilia formation by acting as a selective autophagy adaptor for MYH9/myosin IIA

Yasuhiro Yamamoto [1,2], Haruka Chino[1], Satoshi Tsukamoto [3], Koji L. Ode[4], Hiroki R. Ueda [4,5] & Noboru Mizushima [1✉]

Autophagy regulates primary cilia formation, but the underlying mechanism is not fully understood. In this study, we identify NIMA-related kinase 9 (NEK9) as a GABARAPs-interacting protein and find that NEK9 and its LC3-interacting region (LIR) are required for primary cilia formation. Mutation in the LIR of NEK9 in mice also impairs in vivo cilia formation in the kidneys. Mechanistically, NEK9 interacts with MYH9 (also known as myosin IIA), which has been implicated in inhibiting ciliogenesis through stabilization of the actin network. MYH9 accumulates in NEK9 LIR mutant cells and mice, and depletion of MYH9 restores ciliogenesis in NEK9 LIR mutant cells. These results suggest that NEK9 regulates ciliogenesis by acting as an autophagy adaptor for MYH9. Given that the LIR in NEK9 is conserved only in land vertebrates, the acquisition of the autophagic regulation of the NEK9–MYH9 axis in ciliogenesis may have possible adaptive implications for terrestrial life.

[1] Department of Biochemistry and Molecular Biology, The University of Tokyo, Tokyo, Japan. [2] Department of Respiratory Medicine, Graduate School of Medicine, The University of Tokyo, Tokyo, Japan. [3] Laboratory Animal and Genome Sciences Section, National Institute of Radiological Sciences, National Institutes for Quantum and Radiological Science and Technology, Chiba, Japan. [4] Department of Systems Pharmacology, Graduate School of Medicine, The University of Tokyo, Tokyo, Japan. [5] Laboratory for Synthetic Biology, RIKEN Center for Biosystems Dynamics Research, Osaka, Japan. ✉email: nmizu@m.u-tokyo.ac.jp

The primary cilium is a highly dynamic microtubule-based organelle that protrudes from the plasma membrane when cells exit the cell cycle[1]. It extends from the basal body that matures from the mother centriole of the centrosome. Primary cilia sense and transduce various extracellular stimuli, such as signaling molecules (e.g., Hedgehog, Wnt, Notch, growth factors, and hormones), mechanical forces (e.g., fluid flow and tissue deformation), and environmental cues (e.g., light and odorants), depending on the cell type[2–4]. They regulate diverse developmental and physiological processes, such as embryonic patterning, organogenesis, tissue homeostasis, and cell differentiation[2,5,6]. Accordingly, defects in ciliogenesis lead to a collection of genetic syndromes known as ciliopathies, which have characteristic features that include renal, hepatic, and pancreatic cysts, skeletal anomalies, retinal degeneration, hearing loss, obesity, brain malformations, and mental retardation[7,8].

Macroautophagy (hereafter autophagy) is an intracellular degradation process in which cytoplasmic material is degraded in the lysosome. During autophagy, a fraction of the cytoplasm is sequestered by small membrane cisternae called isolation membranes (or phagophores) to form autophagosomes[9,10]. Autophagosomes then fuse with lysosomes to degrade their engulfed material. Autophagy degrades cytoplasmic contents non-selectively or selectively[11,12]. Selective autophagy cargos include certain soluble proteins, protein aggregates, organelles, including mitochondria, the endoplasmic reticulum (ER), and lysosomes, and intracellular pathogens[13–17]. Selective autophagy is important for maintaining cellular homeostasis and has been implicated in human diseases[18,19]. In mammals, selective autophagy cargos are recognized by ATG8 proteins, which are classified into two subfamilies, namely, the LC3 (including LC3A, LC3B, and LC3C) and GABARAP (including GABARAP, GABARAPL1, and GABARAPL2) subfamilies. They are covalently conjugated to phosphatidylethanolamine in the autophagic membrane and bind to selective cargos with a LC3-interacting region (LIR) motif[20,21]. Alternatively, some LIR-containing soluble proteins, such as SQSTM1 (p62), NBR1, NDP52, OPTN, and TAX1BP1, work as selective autophagy adaptors to mediate binding between ATG8s and cargos[11,12].

Recent evidence suggests that the relationship between autophagy and ciliogenesis is bidirectional; cilia regulate autophagy induction, whereas autophagy regulates ciliogenesis[22,23]. Centriolar satellites are non-membranous organelles in the vicinity of the centrosome, which act as conduits for centrosomal components and regulate ciliogenesis and ciliary functions[24,25]. Centriolar oral-facial-digital syndrome 1 (OFD1) is essential for ciliogenesis; it promotes the docking of the basal body to the plasma membrane and recruits essential ciliary components to the basal body[26,27]. By contrast, OFD1 at centriolar satellites acts as a suppressor of ciliogenesis by blocking recruitment of Bardet–Biedl syndrome 4 (BBS4), a crucial component for cilia elongation. Autophagy degrades OFD1 at centriolar satellites but not at centrioles, thus promoting ciliogenesis[28]. Upregulation of autophagy consistently promotes ciliogenesis in cells across various culture conditions[28–32]. Primary cilia formation is impaired in autophagy-deficient Atg7-KO kidney proximal tubular cells in mice, but the underlying mechanisms remain undetermined[31]. By contrast, other reports have shown that basal autophagy can negatively regulate ciliogenesis[33,34]. Under nutrient-rich conditions, basal autophagy degrades IFT20, a protein essential for ciliogenesis, thus preventing unwanted ciliogenesis during cell proliferation[33]. This complicated relationship between autophagy and cilia may indicate the existence of an unidentified key regulator[22].

Here, we describe the identification of NIMA-related kinase 9 (NEK9) as a GABARAPs-interacting protein. This GABARAP–NEK9 interaction was found to be important for primary cilia formation in culture cells and mice. NEK9 functions as a selective autophagy adaptor for MYH9 (also known as myosin IIA), a negative regulator of ciliogenesis. NEK9-mediated autophagic degradation of MYH9 facilitates actin remodeling and induces ciliogenesis by increasing actin dynamics. We also show that autophagy promotes ciliogenesis primarily by degrading NEK9–MYH9 and OFD1.

## Results

**Differential interactome screen identified NEK9 as a GABARAPs-interacting protein.** To identify substrates or adaptors of selective autophagy, we performed a differential interactome screen using wild-type GABARAPL1 and the LIR-docking site mutant GABARAPL1$^{Y49A/L50A}$, as we previously did with LC3B[35]. The immunoprecipitates were subjected to mass spectrometry (Fig. 1a), and 3129 proteins were detected, including known GABARAP- and LC3-interacting proteins such as p62/SQSTM1, TEX264, and PCM1[12,35,36]. Based on binding intensity and specificity to wild-type GABARAPL1 (Fig. 1b), we focused on NEK9, because it was less characterized in the context of autophagy. NEK9 was also previously identified as a protein interacting with ATG8 family proteins[37].

NEK9 belongs to the NEK family, which is associated mainly with cell-cycle-related functions during mitosis[38,39]. NEK9 is activated during mitosis and phosphorylates various downstream substrates to facilitate proper mitosis progression[40–43]. Homozygous Nek9 knockout mice are embryonic lethal (MGI: 2387995). Some NEK family proteins, such as NEK1, NEK8, and NEK10, have cilia-related functions besides mitotic regulation and are causative genes of human ciliopathies[44–46]. A study of one pedigree indicated that a recessive loss-of-function mutation in NEK9 (a missense mutation in the middle region) causes a lethal skeletal dysplasia (lethal congenital contracture syndrome 10; OMIM 609798), and patient fibroblasts showed a defect in primary cilia formation[47]. NEK8, a close homolog of NEK9, is also a gene responsible for ciliopathy[45]. Very recently, it was reported that NEK9 has a LIR and suppresses selective autophagy by phosphorylating Thr50 within the LIR-docking site of LC3B[48]. However, how NEK9 is involved in primary cilia formation and whether its function relates to autophagy remain unknown.

NEK9 consists of an N-terminal kinase domain, followed by autoinhibitory regulator of chromosome condensation 1 (RCC1)-repeats and a coiled-coil domain[40]. The PSIPRED protein sequence analysis predicted that NEK9 has intrinsically disordered regions (residues 750–891 and 940–979) in the C-terminal region (Fig. 1c). NEK9 interacted with the GABARAP subfamily, but also with the LC3 subfamily to a lesser extent (Fig. 1d). The iLIR search predicted the presence of a LIR (WCLL) in the C-terminal disordered region (Fig. 1c)[49]. Substitution of all four of these residues or even one of the hydrophobic residues (W967 or L970) in the LIR with alanine completely abolished the interaction with GABARAP (Fig. 1e). These results suggest that the LIR is functional and mediates the NEK9–GABARAP interaction, consistent with a recent report[48]. Although the overall structure of NEK9 is conserved among all vertebrates, the LIR is present only in land-living vertebrates, including mammals, reptiles, birds, and amphibians, but not in fish. In addition, the LIR is conserved in the coelacanth (Latimeria chalumnae), an extant species of ancestral lobe-finned fish from which terrestrial vertebrates evolved[50], suggesting that NEK9's LIR-dependent function is important for terrestrial life (Fig. 1c).

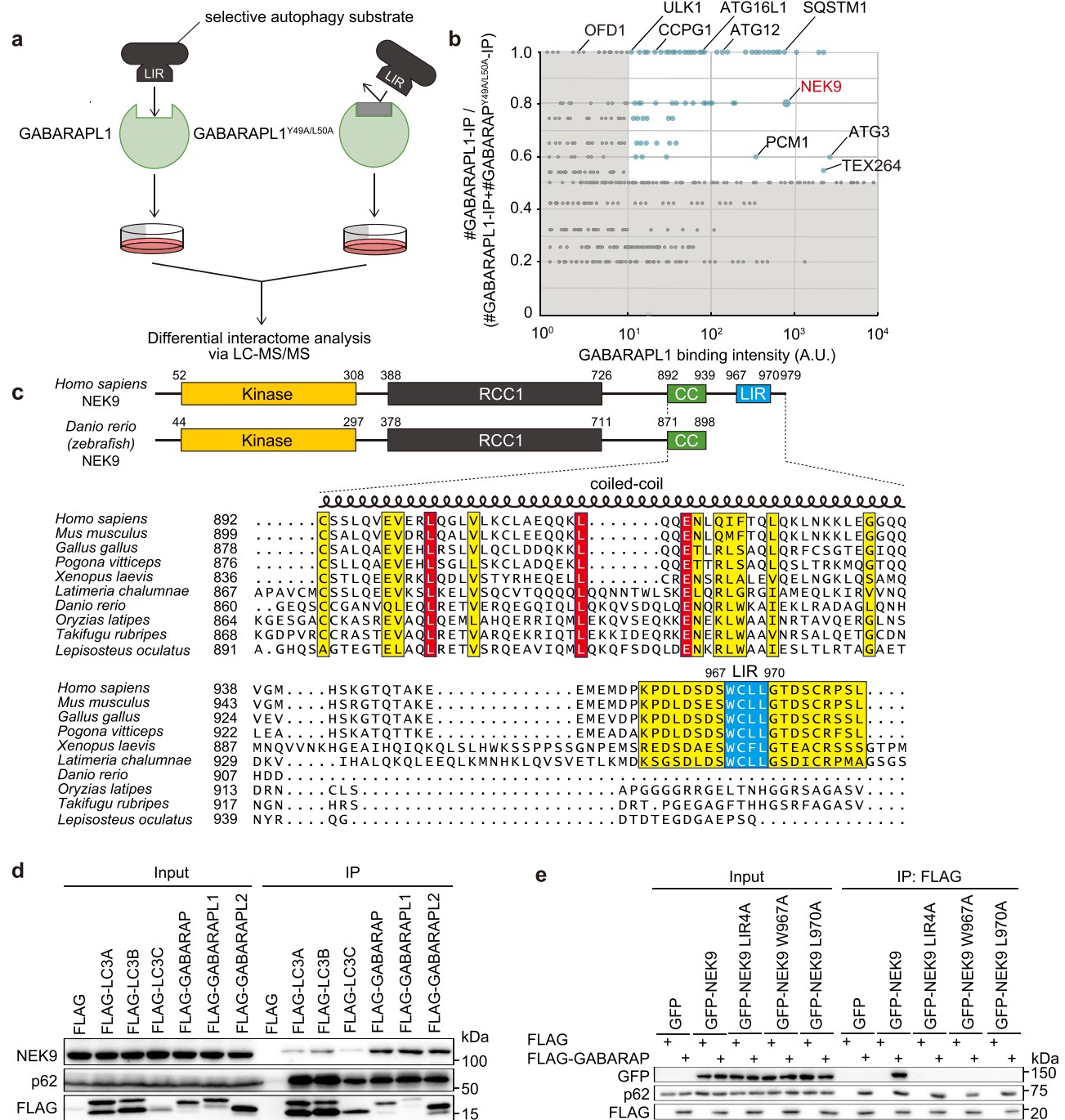

**Fig. 1 Differential interactome screen identified NEK9 as a GABARAP-interacting protein. a** Scheme of the differential interactome screen to identify substrates or adaptors of selective autophagy using GABARAPL1 and its LIR docking site mutant GABARAPL1$^{Y49A/L50A}$. **b** Results of the differential interactome screen. Four independent immunoprecipitation and mass spectrometry (MS) analyses were conducted. The number of times each protein was detected is shown as #GABARAPL1-IP or #GABARAPL1$^{Y49A/L50A}$-IP. The x- and y-axes represent GABARAPL1 binding intensity and the #GABARAPL1-IP / (#GABARAPL1-IP + #GABARAPL1 $^{Y49A/L50A}$ -IP) ratio, respectively. The area defined by x < 10 or y < 0.5 is colored gray. See also Supplementary Data 1. **c** Structures of *Homo sapiens* and *Danio rerio* NEK9 and a multiple sequence alignment of NEK9 proteins in vertebrates. Identical and similar residues are colored in red and yellow, respectively. LIR was predicted by iLIR search. KD, kinase-domain; RCC1, RCC1-repeats; CC, coiled-coil. **d** Immunoprecipitation of FLAG-ATG8s in HEK293T cells. **e** Co-immunoprecipitation of FLAG-GABARAP and wild-type or mutant GFP-NEK9 in HEK293T cells. In GFP-NEK9 LIR4A, the LIR residues (WCLL) were substituted by four alanines. Data are representative of three independent experiments in (**d**) and (**e**).

**NEK9 is degraded by selective autophagy.** To investigate the subcellular localization of NEK9, we observed GFP-tagged NEK9 in mouse embryonic fibroblasts (MEFs). Although diffusely distributed in the cytoplasm under nutrient-rich conditions, NEK9 formed punctate structures under autophagy-inducing starvation conditions (Fig. 2a, b). NEK9 colocalized with the autophagic membrane marker mRuby3-GABARAP and endogenous LC3 and GABARAP (Fig. 2a; Supplementary Fig. 1a–c), as well as the isolation membrane markers FIP200 and WIPI2 (Supplementary Fig. 1d–f)[9,10]. In contrast, NEK9 did not form punctate structures

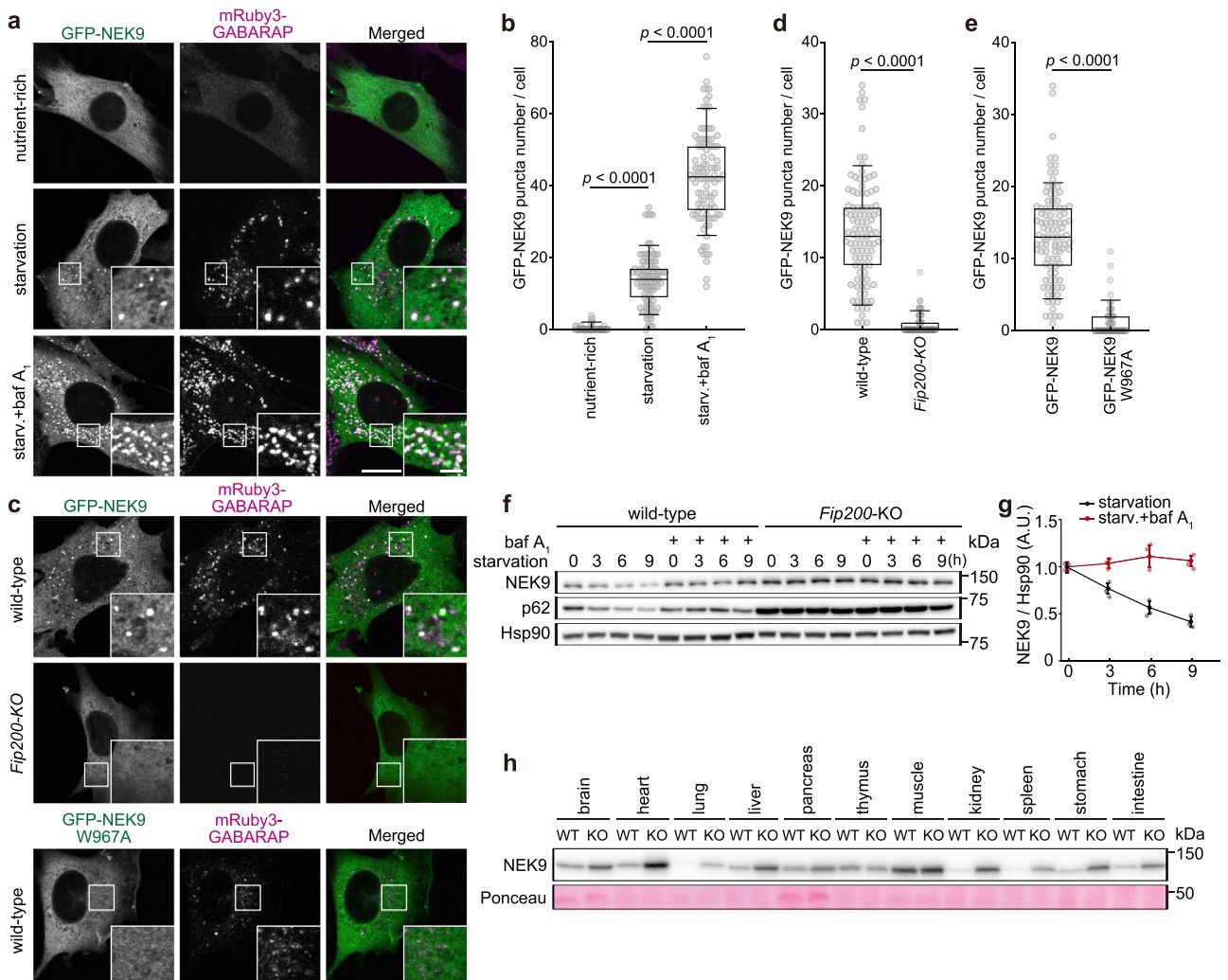

**Fig. 2 NEK9 is degraded by selective autophagy. a** Immunofluorescence microscopy of MEFs expressing GFP-NEK9 and mRuby3-GABARAP under nutrient-rich conditions and amino acid and serum starvation (2 h) conditions with or without 100 nM bafilomycin $A_1$ (baf $A_1$). **b** Quantification of the number of NEK9 puncta in (**a**); $p$ values correspond to a Tukey's multiple comparisons test. **c** Immunofluorescence microscopy of wild-type and *Fip200*-KO MEFs expressing GFP-NEK9 (top), and wild-type MEFs expressing GFP-NEK9 W967A (LIR-mutant) (bottom) after starvation (2 h). **d, e** Quantification of the number of NEK9 puncta in (**c**); $p$ values correspond to two-tailed Mann–Whitney tests. **f** Wild-type and *Fip200*-KO MEFs were incubated under starvation conditions with or without 100 nM bafilomycin $A_1$ for the indicated time. Whole-cell lysates were subjected to immunoblotting. **g** Quantification of the intensity of the NEK9 bands in (**f**). Data represent the mean ± SEM values of three independent experiments. **h** Immunoblotting of indicated organs of three-month-old $Atg5^{+/+}$ (WT) and $Atg5^{-/-}$;*NSE-Atg5* (KO) mice. Data are representative of three biologically independent replicates. For (**b, d,** and **e**), data were collected from 100 cells for each condition. Solid bars indicate the medians, boxes the interquartile range (25th to 75th percentile), and whiskers the 10th to 90th percentile. Scale bars, 10 μm and 3 μm (insets).

in autophagy-deficient *Fip200*-KO cells (Fig. 2c, d) and *Atg3*-KO cells (Supplementary Fig. 1g, h). A LIR mutant of NEK9 (NEK9 W967A) did not localize to the autophagic membrane, even during starvation (Fig. 2c, e). Thus, NEK9 is associating with the autophagic membrane from an early phase in a LIR-dependent manner.

The number of NEK9 puncta increased in the presence of the vacuolar ATPase inhibitor bafilomycin $A_1$ (Fig. 2a, b; Supplementary Fig. 1a, b). Also, NEK9 partially colocalized with lysosome marker LAMP1 under starvation conditions (Supplementary Fig. 1d–f). These data indicate that NEK9 is delivered to lysosomes via autophagy. The level of NEK9 decreased over time during starvation, but this reduction was abolished by bafilomycin $A_1$ (Fig. 2f, g). In contrast, the level of NEK9 did not change during starvation in *Fip200*-KO cells. Furthermore, in $Atg5^{-/-}$; *NSE-Atg5* mice, in which autophagy is blocked in all organs except neuronal cells[51], NEK9 accumulated in most organs

examined (Fig. 2h). These data suggest that NEK9 is degraded by selective autophagy in culture cells and mouse tissues.

A recent study proposed that NEK9 suppresses selective autophagy by phosphorylating Thr50 within the LIR-docking site of LC3B; knockdown or knockout of NEK9 enhanced p62 degradation, whereas NEK9-mediated phosphorylation of LC3B suppressed p62 degradation[48]. However, when we depleted NEK9 using three different siNEK9 sequences, we did not observe enhanced p62 degradation (Supplementary Fig. 2a, b). We also generated *Nek9*-KO cells and measured the level of p62 under nutrient-rich and starvation conditions both with and without bafilomycin $A_1$. Nevertheless, there was no significant difference in p62 level among wild-type cells, *Nek9*-KO cells, and *Nek9*-KO cells re-expressing wild-type NEK9 or a kinase-dead mutant (NEK9 T210A) (Supplementary Fig. 2c, d)[43]. Collectively, we could not observe the inhibitory role of NEK9 in

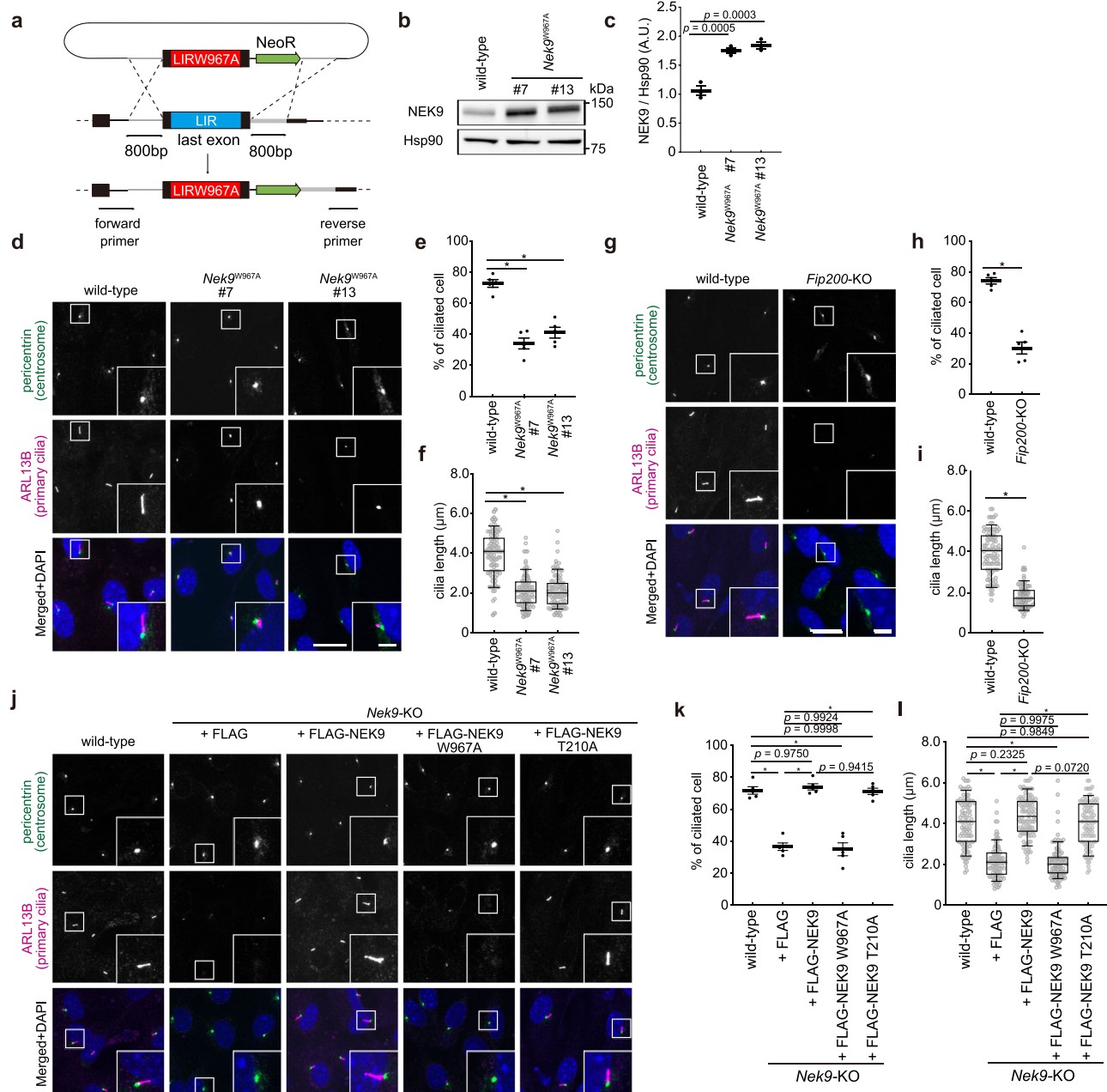

**Fig. 3 Selective autophagy of NEK9 is required for primary cilia formation. a** Generation of homozygous *Nek9*W967A (LIR-mutant) cell lines by CRISPR-mediated recombination using a donor plasmid harboring short homology arms. **b** Immunoblotting of wild-type or *Nek9*W967A MEFs (two independent clones, #7 and #13) cultured in nutrient-rich medium. **c**, Quantification of the intensity of the NEK9 bands in (**b**). Data represent the mean ± SEM of three independent experiments. **d** Immunofluorescence microscopy of wild-type or *Nek9*W967A MEFs after serum starvation (24 h). Centrosomes and primary cilia were stained with anti-pericentrin and anti-ARL13B antibodies, respectively. **e** The frequency of ciliated cells in (**d**). **f** Quantification of cilia length in (**d**). Data were collected from 100 ciliated cells for each cell-type. **g** Immunofluorescence microscopy of wild-type or *Fip200*-KO MEFs after serum starvation (24 h). **h** The frequency of ciliated cells in (**g**), as in (**e**). **i** Quantification of cilia length in (**g**), as in (**f**). Data were collected from 100 ciliated cells for each cell-type. **j** Immunofluorescence microscopy of wild-type or *Nek9*-KO MEFs stably expressing the indicated constructs. **k** The frequency of ciliated cells in (**j**), as in (**e**). **l** Quantification of cilia length in (**j**), as in (**f**). Data were collected from 100 ciliated cells for each cell-type. *p* values correspond to Tukey's multiple comparisons tests in (**c**, **e**, **f**, **k**, and **l**) and two-tailed Mann–Whitney tests in (**h**) and (**i**); *$p < 0.0001$. Scale bars, 10 μm and 3 μm (insets). Data represent the mean ± SEM of five independent experiments (300 cells were counted in each experiment) in (**e**, **h**, **k**). Solid bars indicate the medians, boxes the interquartile range (25th–75th percentile), and whiskers the 10th–90th percentile in (**f**, **i**, and **l**).

selective autophagy of p62 in our experimental settings; instead, our results indicate that NEK9 is degraded by selective autophagy.

**Selective autophagy of NEK9 is required for primary cilia formation.** To investigate NEK9's LIR-dependent function

without affecting its kinase activity, we established NEK9 LIR-mutated MEF clones by CRISPR-mediated recombination using a donor plasmid harboring short homology arms (Fig. 3a; Supplementary Fig. 3a)[52]. In these cells, the endogenous W967 residue, which is essential for binding to GABARAPs (Fig. 1e), was homozygously mutated. NEK9 accumulated in *Nek9*W967A

MEFs (Fig. 3b, c). In these cells, NEK9 did not localize to autophagic membranes (Supplementary Fig. 3b, c) and was not degraded in response to starvation (Supplementary Fig. 3d, e), suggesting that NEK9 is degraded by selective autophagy in a LIR-dependent manner.

We evaluated general autophagic flux using the quantitative GFP-LC3-RFP reporter[53,54], and found that it was not affected in Nek9-KO and Nek9[W967A] cells (Supplementary Fig. 3f). In addition, the lysosomal turnover of p62 and LC3 upon starvation was not impaired in Nek9-KO and Nek9[W967A] cells (Supplementary Fig. 3g, h). Collectively, these results suggest that generalautophagic activity was maintained in Nek9-KO and Nek9[W967A] cells.

Based on a previous report suggesting that NEK9 contributes to ciliogenesis through an unknown mechanism[47], we examined primary cilia in Nek9[W967A] MEFs. Centrosome and primary cilia were labeled with anti-pericentrin and ARL13B (a ciliary membrane protein) antibodies, respectively. Primary cilia were formed from basal bodies (mature mother centrioles) under serum starvation conditions[1], but their formation was impaired in Nek9[W967A] MEFs; relative to wild-type MEFs, the frequency of ciliated cells was lower, and cilia length was shorter (Fig. 3d–f). To exclude the possibility that this phenotype was cell-type specific, we generated NEK9[W967A] HK-2 cells (human renal proximal tubular cell) and obtained similar results (Supplementary Fig. 3i–m). In our experimental conditions, ciliogenesis was also impaired in autophagy-deficient Fip200-KO or Atg3-KO MEFs (Fig. 3g–i; Supplementary Fig. 3n–p), as previously shown[28–32]. Collectively, these results suggest that LIR-dependent selective autophagy of NEK9 is required for ciliogenesis.

To determine whether the kinase activity of NEK9 is required for ciliogenesis, a kinase-dead mutant (NEK9 T210A) was expressed in Nek9-KO MEFs[43]. Ciliogenesis was impaired in Nek9-KO MEFs, as in Nek9[W967A] MEFs, but it was restored by the expression of the kinase-dead NEK9 T210A but not the LIR-mutant NEK9 W967A (Fig. 3j–l; Supplementary Fig. 3q). Thus, selective autophagy of NEK9 is required for ciliogenesis, but its kinase activity is not.

**Selective autophagy of NEK9 is required for primary cilia formation in mouse kidney.** To examine the physiological significance of selective autophagy of NEK9 in vivo, we generated a mouse strain harboring the W967A mutation in NEK9 (Supplementary Fig. 4a). Although homozygous knockout of Nek9 was embryonic lethal (MGI: 2387995), homozygous Nek9[W967A/W967A] mice were viable, fertile, and of normal size and weight. In Nek9[W967A/W967A] mice, NEK9 accumulated in most organs tested, particularly in the kidney (Fig. 4a), suggesting that NEK9 is degraded by selective autophagy in vivo.

The kidney is one of the most frequently affected organs in ciliopathies with primary cilia dysfunction[2,8]. We examined primary cilia of proximal tubular cells in the cortical region of the kidneys and found that primary cilia formation was impaired in Nek9[W967A/W967A] mice (Fig. 4b–d); relative to wild-type littermates, the frequency of ciliated cells was lower in Nek9[W967A/W967A] mice, and cilia length was also shorter. As previously observed in proximal tubule-specific Atg7-KO mice[31], we also observed impaired ciliogenesis in renal tubular cells of Atg5[−/−];NSE-Atg5 mice (Supplementary Fig. 4b–d). These results suggest that selective autophagy of NEK9 is required for primary cilia formation in vivo.

Previous reports suggest that primary cilia negatively regulate cell size through downregulating mTOR activity, and impaired ciliogenesis results in secondary cellular hypertrophy[55]. In ciliopathy models like Kif3a-deficient mice, enlarged cells were observed in renal tubules[56]. Consistently, Nek9[W967A/W967A] mice showed hypertrophy of renal tubular cells (Fig. 4e, f). Renal cysts were not observed in Nek9[W967A/W967A] mice. Furthermore, we and another group have shown cellular hypertrophy in renal tubular cells of autophagy-deficient mice[51,57]. Given that the magnitude of cellular hypertrophy was similar between Nek9[W967A/W967A] and Atg5[−/−];NSE-Atg5 kidneys (Fig. 4e, f; Supplementary Fig. 4e, f), these results imply that the cellular hypertrophy previously observed in autophagy-deficient kidneys could be largely due to disturbed primary cilia formation resulting from a defect in selective autophagy of NEK9.

**NEK9 is a selective autophagy adaptor for MYH9.** There are two possible explanations for ciliogenesis impairment in Nek9[W967A/W967A] mice: (1) accumulated NEK9 may inhibit ciliogenesis, and (2) NEK9 may act as an autophagy adaptor for a ciliogenesis inhibitory protein. The former is unlikely because ciliogenesis was intact in NEK9-overexpressing cells (Supplementary Fig. 5a–d) and in heterozygous Nek9[WT/W967A] mice, despite the accumulation of NEK9 (Supplementary Fig. 5e–g). In contrast, ciliogenesis was affected in Nek9-KO MEFs (Fig. 3j–l). Thus, we hypothesized that NEK9 functions as a selective autophagy adaptor that binds to and degrades a suppressor of ciliogenesis.

NEK9 did not localize to the cilium or basal body under serum starvation conditions (Supplementary Fig. 5h), suggesting that cilia components are not cargos of NEK9. To identify NEK9-interacting proteins, we performed immunoprecipitation and mass spectrometry analysis using FLAG-NEK9. Among the 43 potential interacting proteins (Fig. 5a), those with more than five-fold enrichment in the FLAG-NEK9 immunoprecipitates relative to the FLAG-only immunoprecipitates were individually tested for actual interaction with NEK9. We found that NEK9 specifically interacted with MYH9, an isoform of non-muscle myosin II (Fig. 5b; Supplementary Fig. 5i)[58]. Recent evidence has shown that primary cilia formation is both actin- and microtubule-dependent[59–61]. When ciliogenesis is initiated during cellular quiescence, dynamic remodeling of actin and microtubule cytoskeletons occurs. This enables subsequent migration of the centrosome toward the apical cell surface, where the mother centriole matures into the basal body and the cilium elongates[59,62]. If branched F-actin is stabilized, it impairs the recruitment of ciliogenesis effectors such as RAB11-positive recycling endosomes to the pericentrosomal preciliary compartment, thereby inhibiting the supply of new membranes and proteins to support cilium growth[59,63–65]. Hence, dynamic actin-network remodeling is required for efficient ciliogenesis.

Non-muscle myosin II is an actin-motor protein that has actin cross-linking and contractile abilities. There are three mammalian isoforms (MYH9, myosin IIA; MYH10, myosin IIB; MYH11, myosin IIC), which have both overlapping and unique properties[66]. Myosin II has a central role in cell adhesion, cell migration, and tissue architecture[58,67,68] and is also involved in primary cilia formation[59,62,69,70]. MYH9 suppresses actin dynamics by stabilizing the actin filament network and is hypothesized to be a negative regulator of ciliogenesis[69]. In contrast, MYH10 promotes ciliogenesis by antagonizing MYH9[69,70].

GFP-tagged MYH9 formed punctate structures in wild-type cells under serum starvation conditions and colocalized with NEK9, LC3, and GABARAP, indicating that MYH9 associates with autophagic membranes (Fig. 5c–e; Supplementary Fig. 5j). In contrast, MYH9 did not form punctate structures in Nek9[W967A]

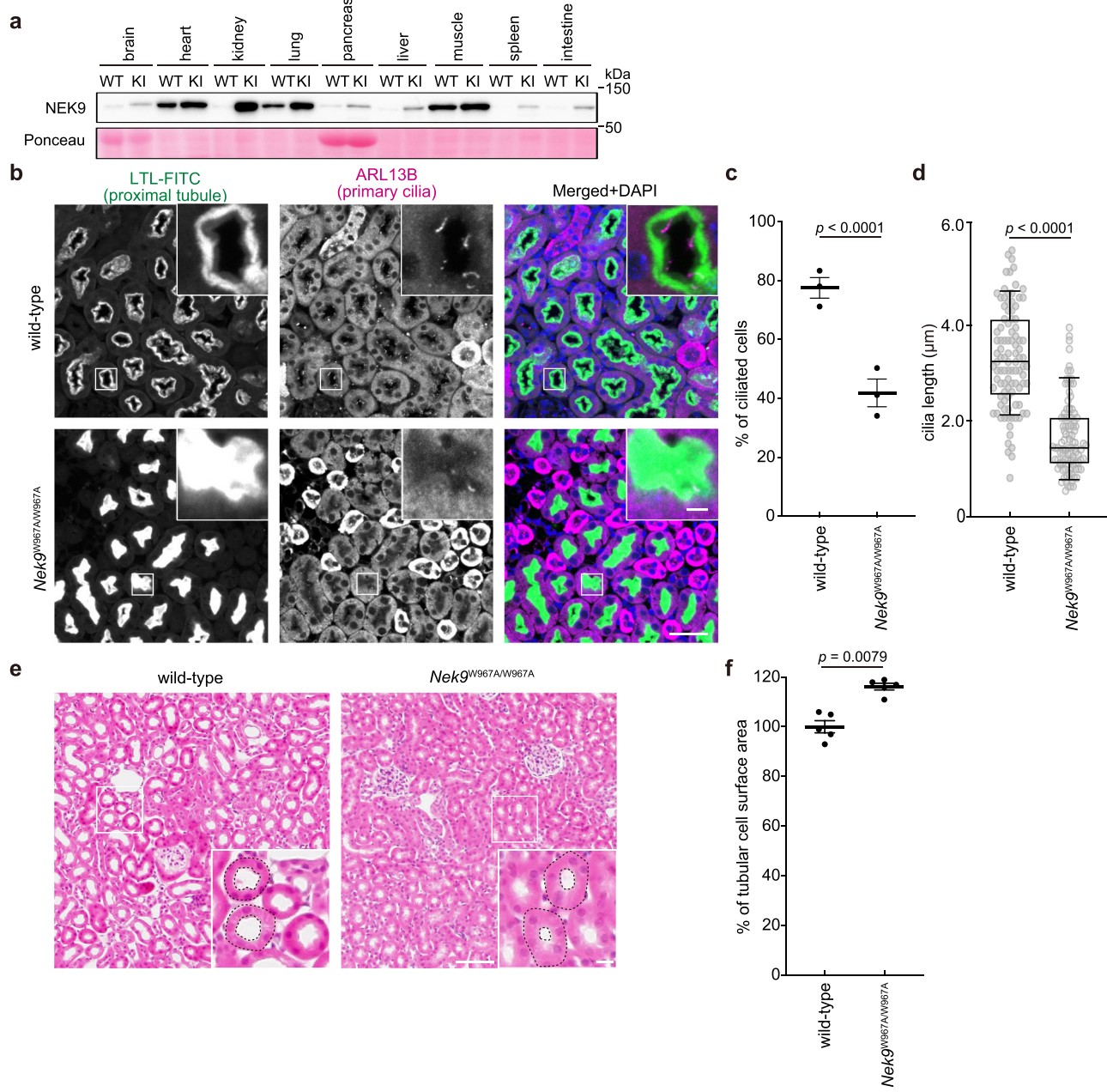

**Fig. 4 Selective autophagy of NEK9 is required for primary cilia formation in mouse kidneys. a** Immunoblotting of the indicated organs of five-month-old wild-type (WT) and *Nek9*^W967A/W967A mice (KI). Data are representative of three biologically independent replicates. **b** Immunohistochemistry of the cortical region of kidneys from five-month-old wild-type and *Nek9*^W967A/W967A mice using LTL-FITC (the lumen of proximal-tubular cells) and anti-ARL13B antibody (primary cilia). Scale bars, 40 µm and 5 µm (insets). **c** Frequency of ciliated cells in LTL-FITC positive cells in (**b**). Data represent the mean ± SEM of three mice (300 cells were counted in each experiment). **d** Quantification of cilia length in LTL-FITC positive cells in (**b**). Data were collected from 100 ciliated cells for each genotype. Solid bars indicate the medians, boxes the interquartile range (25th–75th percentile), and whiskers the 10th–90th percentile. **e** Hematoxylin and eosin staining of the cortical region of kidneys from five-month-old wild-type and *Nek9*^W967A/W967A mice. Scale bars, 100 µm and 10 µm (insets). **f** Measurement of the surface area of tubular cells in (**e**). Examples of measured areas are shown with broken lines in (**e**). Data represent the mean ± SEM of five mice (300 cells were counted in each experiment); *p* values correspond to two-tailed Mann–Whitney tests in (**c**, **d**, and **f**).

cells in which NEK9 is not anchored to autophagic membranes (Fig. 5c–e). Thus, NEK9 recruits MYH9 to autophagic membranes. The interaction between NEK9 and MYH9 became stronger in cilia-inducing serum starvation conditions (Supplementary Fig. 5k, l). MYH9 accumulated in *Nek9*^W967A cells (Fig. 5f, g), *Fip200*-KO cells (Supplementary Fig. 5m, n), and organs of *Nek9*^W967A/W967A mice (Fig. 5h) and *Atg5*^−/−; *NSE-Atg5* mice (except the brain, heart, and muscles, in which

MYH9 is not expressed[71]) (Supplementary Fig. 5o). Although MYH9 and MYH10 are supposed to interact and form hetero-oligomers[66,69], MYH10 did not interact with NEK9 (Fig. 5b) or accumulate in *Nek9*^W967A/W967A or *Atg5*^−/−;*NSE-Atg5* mice (Fig. 5h; Supplementary Fig. 5o). These data suggest that MYH9 is a substrate of selective autophagy that is specifically mediated by NEK9 in a LIR-dependent manner.

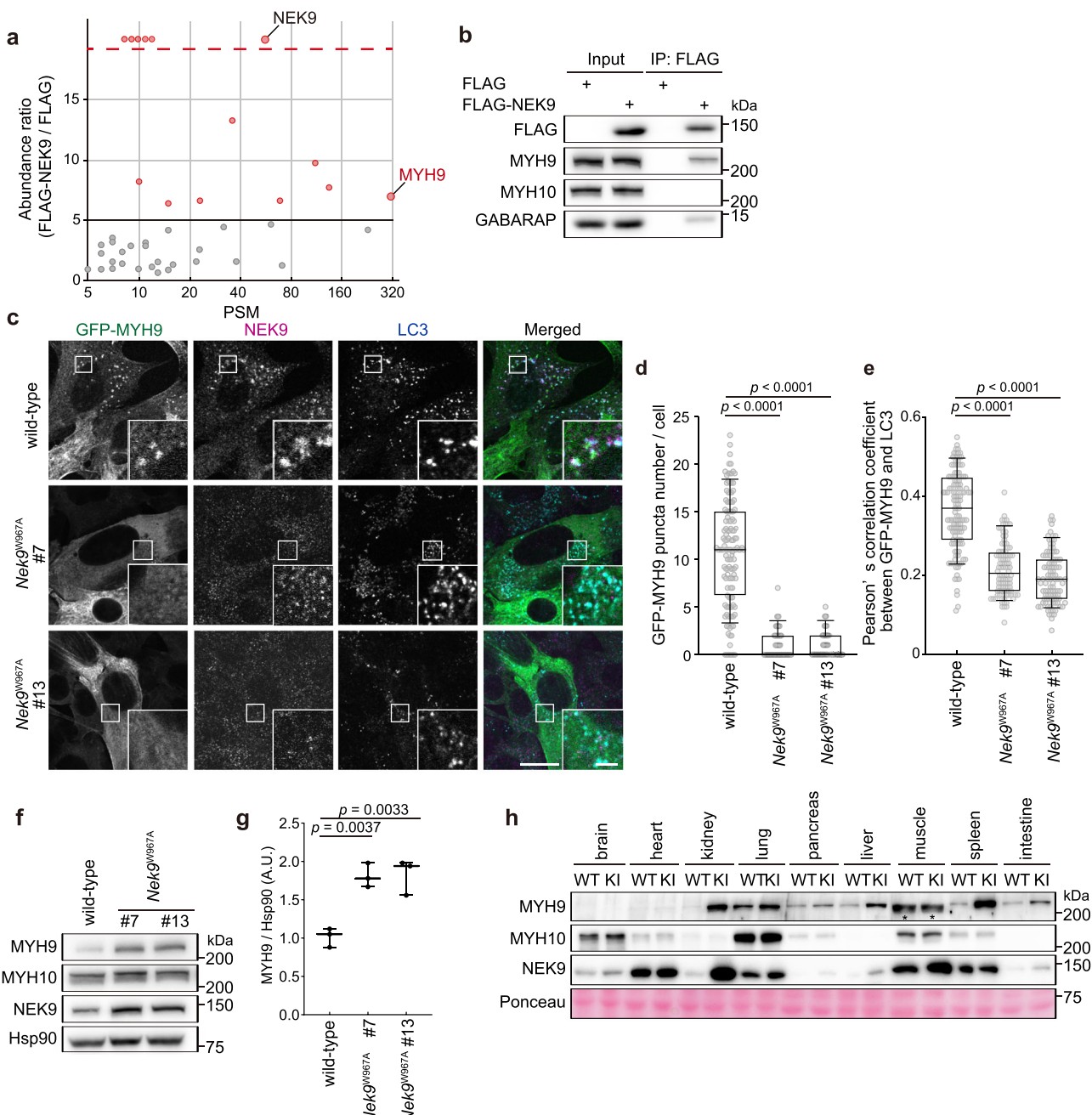

**Fig. 5 NEK9 is a selective autophagy adaptor for MYH9. a** Results of mass spectrometry analysis of FLAG-NEK9 or FLAG immunoprecipitates. The x- and y-axes represent Peptide Spectrum Match (PSM) and abundance ratio (FLAG-NEK9 / FLAG), respectively. Proteins with an abundance ratio above five were tested for actual interaction with NEK9. Proteins above the dotted line were detected only in FLAG-NEK9 immunoprecipitates. See also Supplementary Data 2. **b** Immunoprecipitation using MEFs stably expressing FLAG or FLAG-NEK9 after serum starvation (4 h). Data are representative of three independent experiments. **c** Immunofluorescence microscopy of wild-type and $Nek9^{W967A}$ MEFs stably expressing GFP-MYH9 after serum-starvation (4 h). Cells were stained with anti-NEK9 and anti-LC3 antibodies. Scale bars, 10 μm and 3 μm (insets). **d** Quantification of the number of GFP-MYH9 puncta in (**c**). **e** Colocalization between GFP-MYH9 and endogenous LC3 in (**c**) was determined by calculating Pearson's correlation coefficient between intensities within each cell. Data were collected from 100 cells for each cell-type in (**d**, **e**). Solid bars indicate the medians, boxes the interquartile range (25th–75th percentile), and whiskers the 10th to 90th percentile. **f** Immunoblotting of wild-type and $Nek9^{W967A}$ MEFs. **g** Quantification of the intensity of the MYH9 bands in (**f**). Data represent the mean ± SEM of three independent experiments. **h** Immunoblotting of the indicated organs of five-month-old wild-type (WT) and $Nek9^{W967A/W967A}$ mice (KI). Asterisks (*) indicate non-specific bands in skeletal muscles. Data are representative of three biologically independent replicates; p values correspond to Tukey's multiple comparisons tests.

**NEK9-mediated selective autophagy of MYH9 is required for primary cilia formation**. When MYH9 was overexpressed in wild-type MEFs, primary cilia formation was impaired (Supplementary Fig. 6a–d), confirming a previous report that MYH9 is a suppressor of ciliogenesis[69]. Next, we generated truncated NEK9

constructs to find an MYH9-binding region (Fig. 6a). Mutants lacking the C-terminal region downstream of the LIR (residues 973–979) did not bind to MYH9 (Fig. 6b), suggesting that the region comprising residues 973–979 is important for MYH9 binding. Like the LIR in NEK9, this region is conserved only

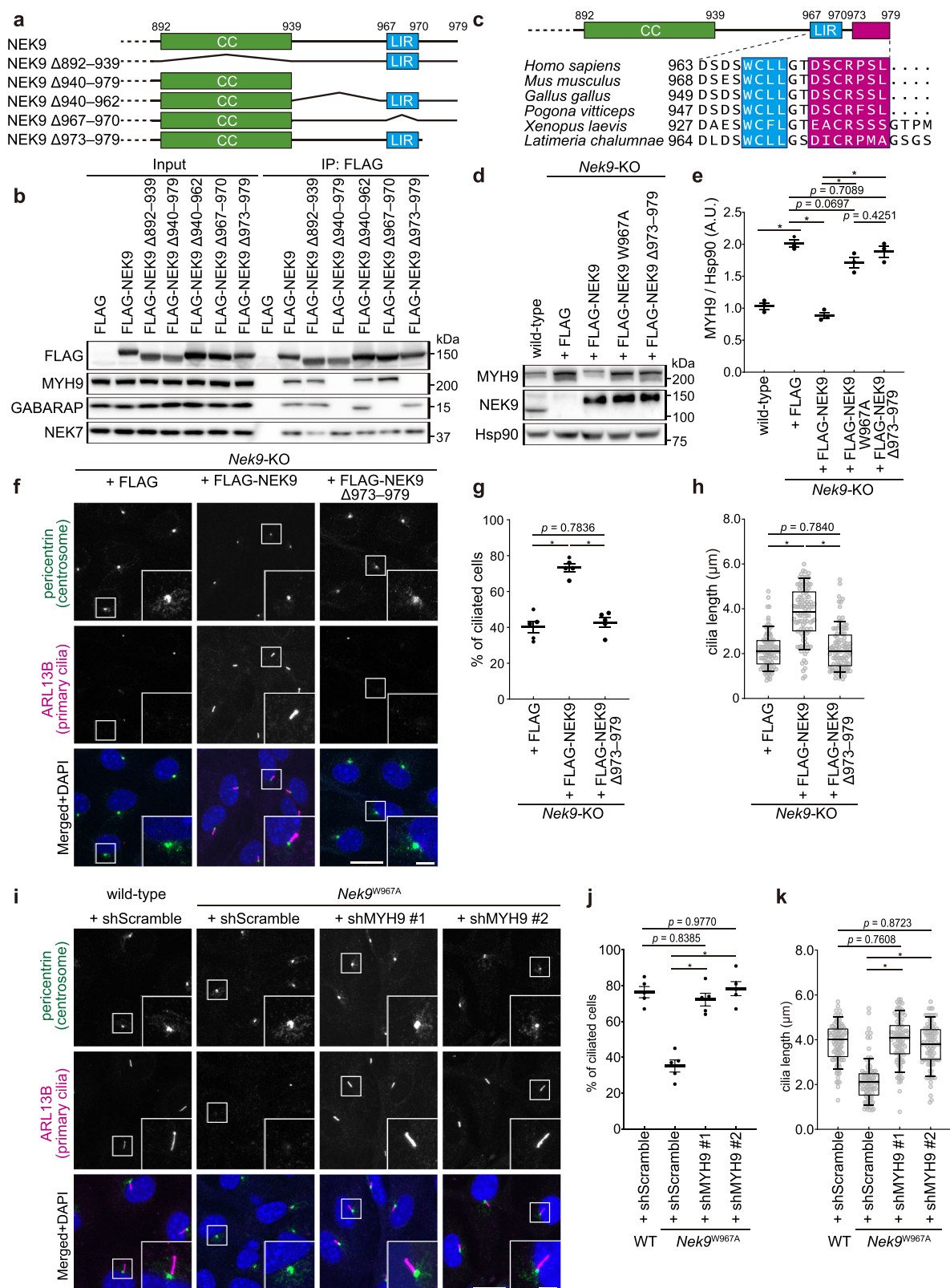

among terrestrial vertebrates (Fig. 6c; see Fig. 1c). NEK8, a close homolog of NEK9 that lacks the corresponding C-terminal region downstream of RCC1-repeats, did not interact with MYH9 (Supplementary Fig. 5i)[72].

To determine whether the LIR and residues 973–979 in NEK9 are required for selective autophagy of MYH9, we expressed the LIR-mutant NEK9 W967A or NEK9 Δ973–979 in Nek9-KO

MEFs. Whereas the expression of wild-type NEK9 abolished the accumulation of MYH9 in Nek9-KO MEFs, the expression of NEK9 W967A or NEK9 Δ973–979 did not, although NEK9 Δ973–979 did localize to autophagosomes, as observed for wild-type NEK9 (Fig. 6d, e; Supplementary Fig. 6e). Thus, both the LIR and residues 973–979 in NEK9 are essential for mediating selective autophagy of MYH9. Accordingly, the

**Fig. 6 NEK9-mediated selective autophagy of MYH9 is required for primary cilia formation. a** Schematic representation of the C-terminal regions of *Homo sapiens* NEK9 and deletion mutants. CC, coiled-coil. **b** Immunoprecipitation using MEFs stably expressing wild-type or deletion mutation NEK9 constructs after serum starvation (4 h). Data are representative of three independent experiments. **c** The C-terminal region of *Homo sapiens* NEK9. The putative MYH9-binding region is colored in magenta (top). Multiple sequence alignment of NEK9 from terrestrial vertebrates (bottom). See also Fig. 1c. **d** Immunoblotting of wild-type or *Nek9*-KO MEFs stably expressing indicated constructs. **e**, Quantification of the intensity of the MYH9 bands in (**d**). Data represent the mean ± SEM of three independent experiments. **f** Immunofluorescence microscopy of *Nek9*-KO MEFs stably expressing indicated constructs after serum starvation (24 h). **g** Frequency of ciliated cells in **f**. Data represent the mean ± SEM of five independent experiments (300 cells were counted in each experiment). **h** Quantification of cilia length in (**f**). Data were collected from 100 ciliated cells for each cell-type. **i** Immunofluorescence microscopy of *Nek9*[W967A] MEFs in which MYH9 was depleted by two independent shRNAs (#1 and #2). See Supplementary Fig. 6e, f for the knockdown efficiency of MYH9 in these cells. **j** Frequency of ciliated cells in (**i**), as in (**g**). Data represent the mean ± SEM of five independent experiments (300 cells were counted in each experiment). **k** Quantification of cilia length in (**i**), as in (**h**). Data were collected from 100 ciliated cells for each cell-type; *p* values correspond to a Tukey's multiple comparisons test; *$p$ < 0.0001. Scale bars, 10 µm and 3 µm (insets). Solid bars indicate the medians, boxes the interquartile range (25th–75th percentile), and whiskers the 10th–90th percentile in (**h**, **k**).

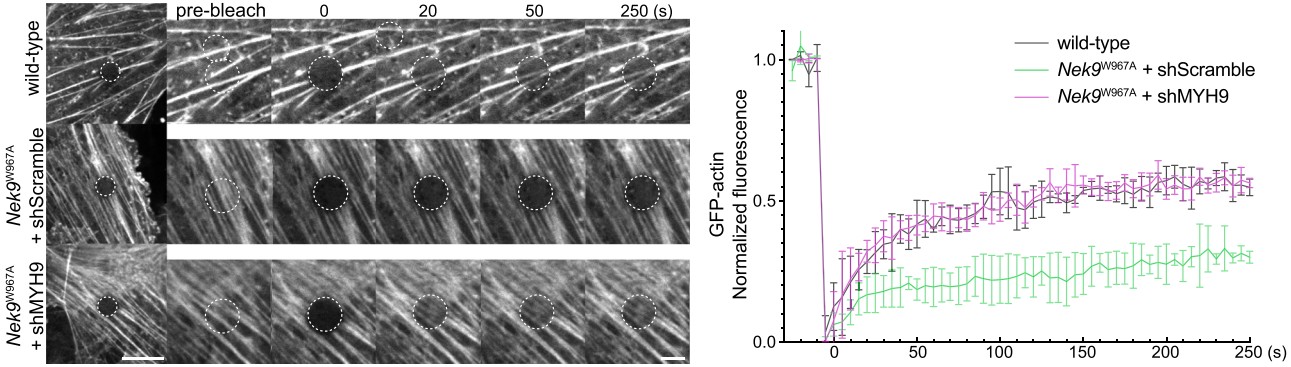

**Fig. 7 Selective autophagy of MYH9 promotes ciliogenesis by increasing actin dynamics.** FRAP analysis of GFP-actin in wild-type or *Nek9*[W967A] MEFs after serum starvation (24 h). MYH9 was depleted by shRNA-mediated knockdown (#1). Images were recorded at 5-s intervals following photobleaching of the indicated area, and fluorescence recovery at different time points was quantified. Data represent the mean ± SEM of 10 cells. Similar results were obtained using shMYH9 #2 (not shown). See Supplementary Fig. 6e, f for the knockdown efficiency. Scale bars, 10 µm and 3 µm (insets).

expression of wild-type NEK9 recovered ciliogenesis in *Nek9*-KO MEFs, but that of NEK9 Δ973–979 did not (Fig. 6f–h). Thus, NEK9-mediated selective autophagy of MYH9 is required for primary cilia formation.

We next determined whether the C-terminal region of NEK9 including the LIR and residues 973–979 is sufficient to facilitate ciliogenesis. The expression of GFP-tagged NEK9 750–979 restored the defects in ciliogenesis in *Nek9*-KO MEFs (Supplementary Fig. 6f–i), suggesting that the C-terminal region of NEK9 including the LIR and residues 973–979 is sufficient to regulate ciliogenesis.

To rule out the possibility that NEK9 regulates ciliogenesis by binding to proteins other than MYH9, we observed the effect of MYH9 depletion on *Nek9*[W967A] cells. Short hairpin RNA (shRNA)-mediated knockdown suppressed MYH9 expression in *Nek9*[W967A] cells to the level observed in serum-starved wild-type cells (Supplementary Fig. 6j, k). In these cells, the defect in ciliogenesis was completely restored (Fig. 6i–k), suggesting that NEK9 regulates ciliogenesis by degrading MYH9.

**Selective autophagy of MYH9 promotes ciliogenesis by increasing actin dynamics.** Given that MYH9 suppresses actin dynamics by stabilizing the actin filament network[69], we monitored actin dynamics in *Nek9*[W967A] cells by fluorescence recovery after photobleaching (FRAP) analysis. The fluorescence recovery of GFP-actin after photobleaching was delayed in *Nek9*[W967A] cells compared to wild-type cells (Fig. 7) and was completely restored by knockdown of MYH9 (Fig. 7), suggesting that actin dynamics are impaired by the accumulation of MYH9

in *Nek9*[W967A] cells. Thus, selective autophagy of MYH9 via NEK9 promotes ciliogenesis by increasing actin dynamics.

**Autophagic degradation of NEK9–MYH9 and OFD1 is required for primary cilia formation.** OFD1 at centriolar satellites is a negative regulator of ciliogenesis and was previously shown to be degraded by autophagy[28]. We confirmed that OFD1 at centriolar satellites was degraded under serum starvation conditions, and this degradation was suppressed in *Fip200*-KO cells (Supplementary Fig. 7a–d). Furthermore, in OFD1-overexpressing cells, OFD1 accumulated at centriolar satellites, even under serum starvation conditions, and primary cilia formation was impaired (Supplementary Fig. 7e–h). Thus, OFD1 at centriolar satellites is indeed a negative regulator of ciliogenesis and undergoes autophagy-dependent degradation.

Next, we examined the relationship between NEK9–MYH9 and OFD1. OFD1 did not colocalize with MYH9 (Supplementary Fig. 8a) and not interact with NEK9 or MYH9 (Supplementary Figs. 5i, 8b). Whereas OFD1-overexpressing cells showed normal actin dynamics (Fig. 8a), OFD1 accumulated at centriolar satellites in *Nek9*[W967A] cells (Fig. 8b, c; Supplementary Fig. 8c, d), suggesting that NEK9 regulates the degradation or dynamics of OFD1. However, although knockdown of OFD1 depleted OFD1 at centriolar satellites, it did not restore the defect in ciliogenesis in *Nek9*[W967A] cells (Fig. 8b–e; Supplementary Fig. 8c–e), indicating that OFD1 accumulation is not the major cause of the impaired ciliogenesis in *Nek9*[W967A] cells. In contrast, MYH9 knockdown depleted OFD1 at centriolar satellites and recovered ciliogenesis (Fig. 8b–e; Supplementary Fig. 8c–e). These data suggest that,

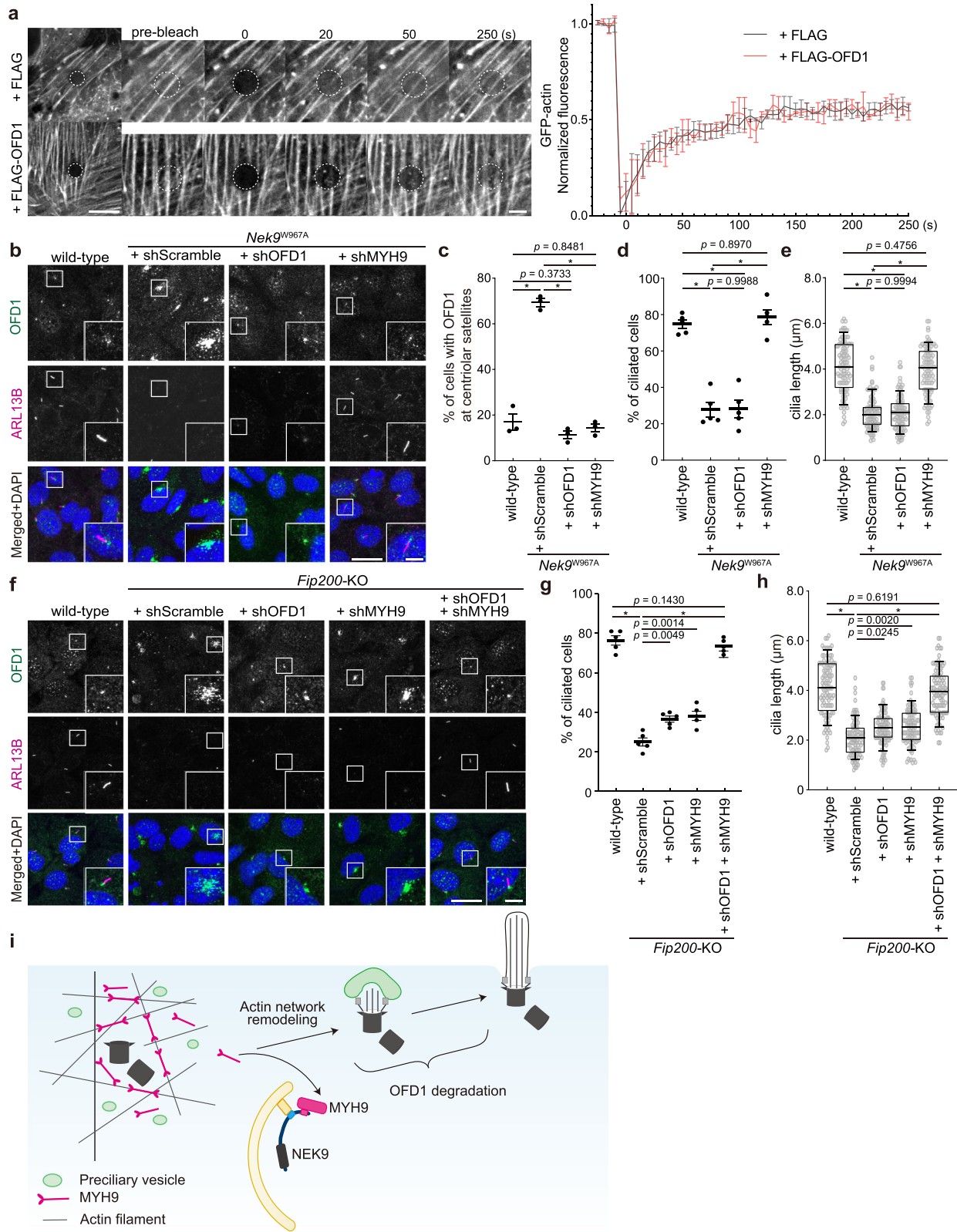

although the degradation of OFD1 depends on MYH9 degradation, it seems to be an indirect effect and that NEK9–MYH9 and OFD1 inhibit ciliogenesis at different steps.

Finally, we determined the contribution of autophagy-dependent degradation of MYH9 and OFD1 to ciliogenesis by depleting MYH9 and/or OFD1 in autophagy-deficient *Fip200*-KO MEFs. Although shMYH9 and shOFD1 suppressed the

expression of these proteins to levels observed in serum-starved wild-type cells (Supplementary Fig. 8f–h), single knockdown of MYH9 or OFD1 only partially rescued the defect in ciliogenesis. In contrast, knockdown of both MYH9 and OFD1 completely restored the defect (Fig. 8f–h), further indicating the indirect relationship between MYH9 and OFD1. These data suggest that the autophagic degradation of both MYH9 and OFD1 is

**Fig. 8 Autophagic degradation of NEK9–MYH9 and OFD1 is required for primary cilia formation. a** FRAP analysis of GFP-actin in wild-type MEFs stably expressing FLAG or FLAG-OFD1 after serum starvation (24 h). Images were recorded at 5-s intervals following photobleaching of the indicated area, and fluorescence recovery at different time points was quantified. Data represent the mean ± SEM of 10 cells. **b** Immunofluorescence microscopy of wild-type or *Nek9*[W967A] MEFs after serum starvation (24 h). OFD1 or MYH9 was depleted by shRNA-mediated knockdown (shMYH9 #1 was used). Similar results were obtained using shMYH9 #2 and two independent shOFD1 (not shown). See Supplementary Fig. 8b–d for the knockdown efficiency. **c** Percentage of cells with centriolar satellites OFD1 in (**b**). Data represent the mean ± SEM of three independent experiments (100 cells were counted in each experiment). **d** Frequency of ciliated cells in (**b**). Data represent the mean ± SEM of five independent experiments (300 cells were counted in each experiment). **e** Quantification of cilia length in (**b**). Data were collected from 100 ciliated cells for each cell-type. Solid bars indicate the medians, boxes the interquartile range (25th–75th percentile), and whiskers the 10th–90th percentile. **f** Immunofluorescence microscopy of wild-type or *Fip200*-KO MEFs after serum starvation (24 h). OFD1 and/or MYH9 were depleted by shRNA-mediated knockdown. See Supplementary Fig. 8e–g for the knockdown efficiency. **g** Frequency of ciliated cells in (**f**), as in (**d**). **h** Quantification of cilia length in (**f**), as in (**e**); *p* values correspond to a Tukey's multiple comparisons test; \**p* < 0.0001. Scale bars, 10 μm and 3 μm (insets). **i** Model of how autophagy drives primary cilia formation.

important for ciliogenesis and that autophagy drives primary cilia formation at least at two distinct steps (Fig. 8i).

## Discussion

Autophagy regulates primary cilia formation, but the underlying mechanism is not fully understood. In this study, we showed that NEK9 functions as a selective autophagy adaptor to degrade MYH9. Considering the previous report that MYH9 suppresses actin dynamics prior to ciliogenesis[69], our data suggested that selective autophagy of NEK9–MYH9 promotes ciliogenesis by increasing actin dynamics. The antagonistic activity between MYH9 and MYH10 indicates that the regulation of relative amounts of these proteins is important for ciliogenesis, but the underlying mechanism has remained largely unknown[69]. Our results suggest that selective autophagy decreases the relative amount of MYH9 to MYH10 and promotes ciliogenesis. The LIR and the putative MYH9 binding region (residues 973–979) of NEK9 are in close proximity. Given that regions around the core LIR could also be involved in interaction with ATG8s, this proximity may prevent NEK9 from binding to both ATG8s and MYH9 at the same time (Fig. 6c)[12,73]. One possible explanation is that NEK9 forms a dimer through the coiled-coil region[38,42] and binds to MYH9 and ATG8s *in trans*.

We showed that OFD1 does not interact with MYH9 and that both associate with autophagic membranes independently (Supplementary Fig. 5i and Supplementary Fig. 8a, b). Consistently, a recent paper revealed that OFD1 has a LIR motif and interacts directly with ATG8 proteins[74]. Nevertheless, our data suggest that the degradation of OFD1 may be partially dependent on that of NEK9–MYH9. Actin network remodeling preceding ciliogenesis allows other organelles, including RAB11-positive recycling endosomes, access to the vicinity of centriolar satellites. Thus, although autophagic degradation of NEK9–MYH9 and OFD1 seems to promote ciliogenesis at different steps, it is possible that increased actin dynamics by selective autophagy of MYH9 indirectly facilitates the degradation of OFD1 at centriolar satellites[59].

This study is consistent with a previous report that NEK9 could be a causative gene of ciliopathies[47]. Lethal skeletal dysplasia and impaired ciliogenesis were observed in a patient with a nonsense mutation (c.1489 C > T; p.Arg497\*) in *NEK9*. No NEK9 protein was detected in patient fibroblasts, indicating both the kinase activity and autophagy adaptor function were lost[47]. Whereas skeletal dysplasia is widely accepted as a common feature of ciliopathy, *Nek9*[W967A/W967A] mice exhibited no skeletal abnormalities, despite showing a defect in ciliogenesis. Therefore, the skeletal dysplasia observed in NEK9-related diseases is probably due to mitotic dysregulation resulting from the loss of NEK9 kinase activity, rather than impaired ciliogenesis. Other NEK proteins, such as NEK1 and NEK8, are also involved in ciliogenesis. However, the NEK9 sequence containing the LIR and

potential MYH9-binding site are not conserved in NEK1 and NEK8. Rather, NEK1 and NEK8 localize directly to the basal body and cilia to promote cilia formation depending on their kinase activity (at least for NEK8)[72]. This is in sharp contrast to the case of NEK9, which promotes ciliogenesis without localizing to these organelles in a kinase-activity-independent manner (Fig. 3j; Supplementary Fig. 6g). Nevertheless, we do not rule out the possibility that the kinase activity of NEK9 contributes to ciliogenesis, as observed for other NEK proteins. It would be reasonable to assume that NEK9 acquired the LIR-dependent function in ciliogenesis because it already had a still-undiscovered cilia-related function.

We did not observe any renal cysts in *Nek9*[W967A/W967A] mice. Consistently, renal cysts have never been reported in autophagy gene-deficient mice[31,51,57]. Although impaired degradation of NEK9–MYH9 by selective autophagy suppressed ciliogenesis and induced secondary cellular hypertrophy in renal tubular cells, cilia managed to grow in about half of these cells (Fig. 4b; Supplementary Fig. 4b), which may prevent cyst formation.

A dramatic functional evolution occurred in the vertebrate kidney during the fish-to-tetrapod transition to overcome physiological changes[75], including acquisition of the ability to excrete nitrogen or maintain homeostasis of water and various small molecules. During this evolution, the vertebrate kidney acquired primary cilia; in contrast to primary cilia found in the kidneys of higher vertebrates, all cilia in fish kidneys are motile[76,77]. Therefore, we speculate that the evolutionary acquisition of the LIR of NEK9 was critical for the newly-acquired primary cilia formation in the tetrapod kidney and thus for the adaptation of vertebrates to terrestrial habitats.

## Methods

**Cell lines.** Mouse embryonic fibroblasts (MEFs), HeLa cells, and HEK293T cells were cultured in Dulbecco's modified Eagle's medium (DMEM) (D6546; Sigma-Aldrich) supplemented with heat-inactivated 10% fetal bovine serum (FBS; 173012; Sigma-Aldrich) and 2 mM L-glutamine (25030-081; GIBCO) in a 5% CO$_2$ incubator. HK-2 cells were cultured in DMEM/F-12 (Dulbecco's modified Eagle's Medium Nutrient Mixture F-12; 11320033; GIBCO) medium supplemented with 10% FBS, 1% ITS (insulin, transferrin and sodium selenite media supplement, I1884; Sigma-Aldrich), and 2 mM L-glutamine. For the starvation treatment, cells were washed twice with phosphate-buffered saline (PBS) and incubated in amino acid-free DMEM (048-33575; Wako Pure Chemical Industries) without serum. For the serum-starvation treatment, cells were washed twice with PBS and incubated in DMEM supplemented with 2 mM L-glutamine. *Fip200*-KO[78] and *Atg3*-KO[79] MEFs have been described previously. To generate stable cell lines, cells were cultured with retrovirus or lentivirus, and stable transformants were selected with puromycin (P8833; Sigma-Aldrich), blasticidin (022-18713; Wako Pure Chemical Industries), or geneticin (10131035; Thermo Fisher Scientific).

**Mouse strains.** All animal experiments were approved by the Institutional Animal Care and Use Committee of the University of Tokyo (Medical-P17-084) and the Animal Care and Use Committee of the National Institute of Quantum and Radiological Science and Technology (1610111 and 1610121). Wild-type C57BL/6 mice were obtained from Japan SLC, Inc. *Atg5*[−/−];*NSE-Atg5* mice have been previously described[51]. There was no bias between male and female mice used in

our study. Wild-type or mutant neonatal mice were prepared by mating male and female mice at 2–10 months of age. We analyzed 5-month-old $Nek9^{W967A/W967A}$ or $Nek9^{WT/W967A}$ mice and three-month-old $Atg5^{-/-}$;NSE-Atg5 mice for the analysis of the kidney. All mice were housed in a specific pathogen-free room maintained at a constant ambient temperature of 22-26 degree Celsius, 40-65% of humidity under a 12 h light/dark cycle with free access to food and drink.

**Generation of $Nek9^{W967A/W967A}$ mice.** $Nek9^{W967A/W967A}$ mice were generated by CRISPR-mediated knockin of the W967A mutation at the mouse Nek9 genomic locus. Oocyte/embryo manipulation was performed as previously described[54]. Briefly, MII oocytes were collected from superovulated C57BL/6 J females and fertilized in vitro. The fertilized one-cell embryos were washed with Opti-MEM I (Life Technologies) and transferred into a 1-mm electroporation cuvette (CUY501P1-1.5; Nepa Gene Co.) containing 5 µL of a mixture of recombinant Cas9 protein (50 ng/µL, Nippon Gene), gRNA (0.8 µM, Alt-R CRISPR-Cas9 System; Integrated DNA Technologies), and ssODN (0.5 µg/µL, PAGE Ultramer DNA Oligo; Integrated DNA Technologies) in Opti-MEM I. The sequence of the gRNA targeting exon 22 of mouse Nek9 was 5′-TCGGAGTCCTGGTGCCTCCT-3′. The sequence of the donor oligonucleotide (ssODN) was 5′-TCCTGAGGGCTATGTG GGCTCAGGAGACTAGAGGCTGGGTCGACAAGAGTCTGTTCCGAGGAGG CAGGCGGACTCCGAGTCTAAGTCAGGCTTTGGATCCATTTCCATTTCTTC CTTTGCTGTCTGGGT-3′. Electroporation was conducted using a NEPA21 Super Electroporator (Nepa Gene Co.) under the following conditions: four poring pulses at 40 V, with a pulse width of 3.5 ms and pulse interval of 50 ms; five transfer pulses at 7 V, with a pulse width of 50 ms and pulse interval of 50 ms. After electroporation, the embryos were immediately recovered from the cuvette, washed with Opti-MEM I, and cultured overnight in KSOM medium. The embryos that developed normally to the two-cell stage were transferred to the oviduct of pseudopregnant ICR females (CLEA Japan, Inc.) on the day of the vaginal plug (Day 0.5). Genomic DNA of offspring (F0 founders) was collected by tail biopsy and used for genotyping. F0 founders harboring potential mutant alleles were bred to wild-type C57BL/6 J mice, and mutations in the F1 generation were confirmed by sequencing. Genotyping was performed by PCR using primers flanking the target site (forward primer, 5′-CAGCCAGTTCACTTTTCATACTACATCCCCCAAA TG-3′; reverse primer, 5′-CAAAGCCAGGTCCACAGGACCCTTCCATTCTC CCA-3′).

**Plasmids.** cDNAs encoding human NEK9 (NP_001316166.1), MYH9 (NP_002464.1), OFD1 (NP_001317138.1), GABARAP (NP_009209), GABAR-APL1 (NP_113600), LC3B (NP_073729), Actin (NP_001092.1) were inserted into pMRX-vector[80]. DNAs encoding enhanced GFP, codon-optimized mRuby3 (modified from pKanCMV-mClover3-mRuby3; 74252; Addgene)[81], and 3 × FLAG were also used for tagging. pMXs-GFP-LC3-RFP vector was described previously[81]. Mutated or truncated constructs were prepared by PCR-mediated site-directed mutagenesis. Single-guide RNAs (sgRNAs) targeting exon 2 of mouse Nek9 (5′-CACCACCCTGCTGATTGAGC-3′, 5′-GCTGATTGAGCTGGAGTACT-3′), exon 22 of mouse Nek9 (5′-TCGGAGTCCTGGTGCCTCCT-3′) and exon 22 of human NEK9 (5′-TACAGGAGTCTGTTCCCAGG-3′) were cloned into pSpCas9(BB)-2AGFP (a gift from Dr. F. Zhang, Broad Institute of Massachusetts Institute of Technology; Addgene #48138). For generation of $Nek9^{W967A}$ MEFs and $NEK9^{W967A}$ HK-2 cells, donor plasmids harboring short homology arms (800 bps each, with one arm covering exon 22 of mouse Nek9 or human NEK9, which includes the bases encoding the LIRs) were generated from the pCMV vector. The homology arms were amplified by PCR and inserted between the BamHI and HindIII sites of the pCMV vector. The primer sequences were 5′-GGATCCGAAG GGAGGGAAGAGTCATTCCTTGGCTCTG-3′, 5′-GGATCCTTTCCTGAGGGC TATGTGGGCTCAGGAGACT-3′, 5′-AAGCTTCTGGGACCCAAAGAACTTCA CCGCACACTTAC-3′, and 5′-AAGCTTCTGAAGCTTCAAACATCCACGGTG AAAGCAAC-3′ for $Nek9^{W967A}$ MEFs and 5′-GGATCCCATTAAACATTTTA GTGAAAATACTTCAAAG-3′, 5′-GGATCCCTAGAGGCTGGGTCTACAGG AGTCTGTTCC-3′, 5′-AAGCTTTCTCCTGAGCCTGTAGAGCCCCCAGGAG ACT-3′, and 5′-AAGCTTCTAGACATTCAGAAAGAAAGTGGTTGGGGCTG-3′ for $NEK9^{W967A}$ HK-2 cells. A neomycin-resistance gene was inserted into the PstI site of the pCMV vector. The W967A mutation in NEK9 was introduced by PCR-mediated site-directed mutagenesis. A complete list of all primers used was supplied in Supplementary Table 3.

**Antibodies and reagents.** The following antibodies were used for immunoblotting: mouse monoclonal antibody against HSP90 (1:5000 dilution, 610419; BD) and rabbit polyclonal antibodies against NEK9 (1:5000 dilution, A301-139A; Bethyl), NEK8 (1:5000 dilution, A0984; ABclonal), NEK7 (1:5000 dilution, 3057 S; Cell Signaling), MYH9 (1:5000 dilution, A0173; ABclonal), MYH10 (1:5000 dilution, A12029; ABclonal), OFD1 (1:2500 dilution, NBP1-89355; Novus Biologicals), GABARAP (1:2500 dilution, 13733 S; Cell Signaling Technology), p62/SQSTM1 (1:5000 dilution, PM045; MBL), GFP (1:10000 dilution, A6455; Thermo Fisher Scientific), and FLAG (1:10000 dilution, F7425; Sigma-Aldrich), LC3 (1:10000 dilution, M152-3; MBL). HRP-conjugated anti-mouse and anti-rabbit IgG (1:10000 dilution, 111-035-003, 111-035-144; Jackson ImmunoResearch Laboratories) antibodies were used as secondary antibodies. The following antibodies were used

for immunocytochemistry and immunohistochemistry: rabbit polyclonal antibodies against FIP200 (1:200 dilution, 17250-1-AP; ProteinTech), WIPI2 (1:200 dilution, SAB4200400; Sigma-Aldrich), LAMP1 (1:200 dilution, ab24170; Abcam), NEK9 (1:200 dilution, A301-139A; Bethyl), OFD1 (1:100 dilution, NBP1-89355; Novus Biologicals), and pericentrin (1:200 dilution, abcam; 4448) and mouse monoclonal antibodies against LC3 (1:100 dilution, CTB-LC3-2-IC; CosmoBio), NEK9 (1:100 dilution, sc-100401; Santa Cruz), FIP200 (1:200 dilution, MABC128; Sigma-Aldrich), WIPI2 (1:200 dilution, MABC91; Sigma-Aldrich), LAMP1 (1:200 dilution, ab25630; abcam), and ARL13B (1:200 dilution, ab136648; abcam). Alexa Fluor 488-conjugated goat anti-mouse IgG, Alexa Fluor 568-conjugated goat anti-rabbit IgG, and Alexa Fluor 660-conjugated goat anti-mouse IgG (1:200 dilution, A-11029, A-11036, and A-21055; Thermo Fisher Scientific) were used as secondary antibodies. Hoechst 33342 (1:200 dilution, H342; Dojindo Molecular Technologies) or DAPI-containing SlowFade Antifade Mountant (S36938; Thermo Fisher Scientific) was used to stain DNA. LTL-FITC (1:200 dilution, Vector Laboratories; FL-1321-2) was used to stain proximal tubular cells. For transient expression, Fugene HD (VPE2311; Promega) was used. For bafilomycin A$_1$ treatment, cells were cultured with 100 nM bafilomycin A$_1$ (B1793; Sigma-Aldrich).

**Preparation of lentivirus and retrovirus.** To prepare the lentivirus, HEK293T cells were transiently transfected with a lentiviral vector together with pCMV-VSV-G (a gift from Dr. R. A. Weinberg, Whitehead Institute for Biomedical Research) and psPAX2 (a gift from Dr. D. Trono, Ecole Polytechnique Federale de Lausanne) using Lipofectamine 2000 (11668019; Thermo Fisher Scientific). After cells were cultured for 2–3 days, the supernatant was collected and passed through a 0.45-mm syringe filter unit (SLHV033RB; EMD Millipore). To prepare the retrovirus, HEK293T cells were transiently transfected with a retroviral vector together with pCG-VSV-G and pCG-gag-pol (a gift from Dr. T. Yasui, National Institutes of Biomedical Innovation, Health and Nutrition) using Lipofectamine 2000, and viral particles were collected from the supernatant as described above.

**Establishment of Nek9-KO MEFs.** Wild-type MEFs were transfected with pSpCas9(BB)-2AGFP encoding sgRNAs targeting exon 2 of mouse Nek9. Two days after transfection, GFP-positive cells were isolated using a cell sorter (MoFlo Astrios EQ; Beckman Coulter), and single clones were obtained. Clones with mutations in both alleles were identified by immunoblotting and sequencing of genomic DNA.

**Establishment of $Nek9^{W967A}$ MEFs and $NEK9^{W967A}$ HK-2 cells.** To generate $Nek9^{W967A}$ MEFs, wild-type MEFs were co-transfected with pSpCas9(BB)-2AGFP encoding the sgRNA targeting exon 22 of mouse Nek9 and the donor plasmid described above. Two days after transfection, GFP-positive cells were isolated using a cell sorter. After geneticin selection, single clones were obtained by the limiting dilution method. Clones harboring the W967A mutation and the neomycin-resistant cassette in both alleles were identified by PCR (forward primer, 5′-CTTTCACCCCTAACGTGAGTTTGGACTTCTTACTTTGTG-3′; reverse primer, 5′-CTGAAGCTTCAAACATCCACGGTGAAAGCAACCTGAGC-3′) and sequencing of genomic DNA. $NEK9^{W967A}$ HK-2 clones were similarly generated and identified by PCR (forward primer, 5′-ATAGACACCTTGTATGGTTCTTT GGAGGATTAAATGAACT-3′; reverse primer, 5′-AAGAAAGTGGTTGGGGCT GCTGATATCAAGATCAGAACC-3′) and sequencing of genomic DNA.

**siRNA-mediated knockdown.** Stealth RNAi oligonucleotides were obtained from Thermo Fisher Scientific. The following sequences were used: siNEK9#1, 5′-AA UAGCAGCUGUGUGAGUCUUG-3′; siNEK9#2, 5′-GCAGCCAAACUUUGAU UAAAGUU-3′; siNEK9#3, 5′-GCUGCCUUGGGAAUUCAGUACCA-3′; and siLuciferase (siLuc), 5-AAUUAAGUCCGCUUCUAAGGUUUCC-3′. The stealth RNAi oligonucleotides were transfected into cells using Lipofectamine RNAiMAX (13778150; Thermo Fisher Scientific) according to the manufacturer's instructions. Cells were harvested three days after transfection.

**shRNA-mediated knockdown.** pLKO.1-blast vectors (a gift from Keith Mostov; Addgene plasmid # 26655) respectively containing shRNAs to Myh9 (sh#1, 5′-CCATACAACAAATACCGCTT-3′; sh#2, 5′-GGTAAATTCATTCGTATCAA-3′) and Ofd1 (sh#1, 5′-GCTAGAATCTTTAGAGACAAA-3′; sh#2, 5′-TCACAAGAA GTCACGTAATAT-3′) as well as a non-targeting control (shScramble, 5′-CAAC AAGATGAAGAGCACCAA-3′) were packaged into the lentivirus as described above. Cells were infected with the lentivirus, and stable transformants were selected with blasticidin. The knockdown efficiency was measured by immunoblotting.

**Immunofluorescence.** Cells grown on coverslips were washed with PBS and fixed in 4% paraformaldehyde (PFA; 09154-85; Nacalai Tesque) for 10 min at room temperature or in 100% methanol (21914-03;Nacalai Tesque) for 5 min at −30 °C. The PFA-fixed cells were permeabilized with 50 mg/mL digitonin (D141; Sigma-Aldrich) in PBS for 5 min, blocked with 3% bovine serum albumin (BSA; 011-27055; Wako Pure Chemical Industries) in PBS for 30 min, and then incubated

with primary antibodies for 1 h at room temperature or overnight at 4 °C (for anti-OFD1 antibody). After washing three times with PBS, cells were incubated with Alexa Fluor 488/568/660-conjugated goat anti-mouse or anti-rabbit IgG secondary antibodies for 1 h at room temperature. Fluorescence microscopy was performed using a confocal laser microscope (FV1000 IX81; Olympus) with a 100 × oil-immersion objective lens (1.40 NA; Olympus) and captured with FluoView software (FV10-ASW, Olympus). The number of punctate structures and the colocalization rate were determined using Fiji software (ImageJ). For primary cilia quantification, cells were incubated to confluency and serum-starved for 24 h before fixation, and immunofluorescence images were obtained in 15 serial z-stack series at 0.4-µm intervals. The percentage of cells with cilia was measured by dividing the number of cilia (ARL13B-positive structures) by the number of nuclei (DAPI). Cilia length was determined by the Pythagorean theorem method[82].

**Histology and immunohistochemistry**. For hematoxylin and eosin staining, mouse tissues were fixed in 4% PFA overnight at 4 °C, and infiltrated with 15% and then 30% sucrose in PBS for 4 h each. Tissues were then embedded in Tissue-Tek OCT Compound (Sakura Japan Co.) and frozen at −80 °C. Sections (7 µm) were prepared using a cryostat (CM3050 S, Leica Microsystems) and mounted on slides. Cryosections were stained with hematoxylin and eosin and photographed using a microscope (BX51; Olympus) equipped with a digital camera (DP70; Olympus). For immunohistochemistry, mouse tissues were frozen after fixation in Tissue-Tek OCT Compound and cryosections were prepared. Cryosections were permeabilized with 50 mg/mL digitonin in PBS for 5 min, blocked with 3% BSA in PBS for 30 min, and incubated with primary antibodies in 3% BSA for 1 h, followed by a PBS wash and incubation for 1 h with secondary antibodies.

**Immunoprecipitation and immunoblotting**. Cells were lysed in a lysis buffer (50 mM Tris-HCl, pH 7.5, 150 mM NaCl, 1 mM EDTA, 1% Triton X-100, PhosSTOP [4906837001; Roche], and complete EDTA-free protease inhibitor [03969-21;Nacalai Tesque]). After centrifugation at $17,700 \times g$ for 10 min, the supernatants were incubated with anti-FLAG M2 affinity gel (A2220; Sigma-Aldrich) for 3 h at 4 °C with gentle rotation. Precipitated immunocomplexes were washed three times in washing buffer (50 mM Tris-HCl, pH 7.5, 150 mM NaCl, 1 mM EDTA, and 1% Triton X-100) and boiled in sample buffer (46.7 mM Tris- HCl, pH 6.8, 5% glycerol, 1.67% sodium dodecyl sulfate, 1.55% dithiothreitol, and 0.02% bromophenol blue). Samples were subsequently separated by sodium dodecyl sulfate polyacrylamide gel electrophoresis and transferred to Immobilon-P polyvinylidene difluoride membranes (IPVH00010; EMD Millipore). Immunoblotting analysis was performed with the indicated antibodies. Super-Signal West Pico Chemiluminescent Substrate (1856135; Thermo Fisher Scientific) or Immobilon Western Chemiluminescent HRP Substrate (P90715; EMD Millipore) was used to visualize the signals, which were detected using the Fusion Solo 7 S system (M&S Instruments). Appropriate contrast and brightness adjustment and quantification were performed using Fiji software (ImageJ).

**Flow cytometry**. Cells stably expressing GFP-LC3-mRFP were treated with 250 nM Torin 1 (4247, Tocris Bioscience) in DMEM for 24 h. The cells were harvested by trypsinization followed by centrifugation at 2,500 g for 2 min, resuspended in ice-cold PBS, treated with 1% 7-amino-actinomycin D (51-68981E, BD Pharmingen) for 10 min on ice, and then diluted with ice-cold PBS. The cells were analyzed by the cell analyzer (EC800, SONY) equipped with 488-nm and 561-nm lasers. Data were processed using a Kaluza Analysis 2.1 software (Beckman Coulter).

**Liquid chromatography-tandem mass spectrometry (LC-MS/MS) analysis of GABARAPL1 and GABARAPL1^Y49A/L50A immunoprecipitates**. FLAG-GABARAPL1 and FLAG-GABARAPL1$^{Y49A/L50A}$ plasmids were prepared as described above. FLAG-GABARAPL1 and FLAG-GABARAPL1$^{Y49A/L50A}$-interacting proteins were immunoprecipitated from HEK293T cells and identified by LC-MS/MS analysis using a high-performance LabDroid system at Robotic Biology Institute (https://rbi.co.jp/en/). Briefly, HEK293T cells transiently expressing FLAG-GABARAPL1 and FLAG-GABARAPL1$^{Y49A/L50A}$ were lysed with lysis buffer (20 mM HEPES-NaOH, pH 7.5, containing 1% digitonin, 150 mM NaCl, 50 mM NaF, 1 mM Na$_3$VO$_4$, 5 mg/mL leupeptin, 5 mg/mL aprotinin, 3 mg/mL pepstatin A, and 1 mM phenylmethylsulfonylfluoride) and centrifuged at $20,000 \times g$ for 10 min to remove insoluble materials. The supernatants were subjected to immunoprecipitation using anti-FLAG M2 magnetic beads (M8823; Sigma-Aldrich). Precipitated immunocomplexes were washed three times in washing buffer (10 mM HEPES-NaOH, pH 7.5, 150 mM NaCl, and 0.1% Triton X-100) and eluted with FLAG peptides. The samples obtained were subjected to trichloroacetic acid precipitation. The resulting pellets were dissolved in 0.1 M ammonium bicarbonate (pH 8.8) containing 7 M guanidine hydrochloride, reduced using 5 mM TCEP (tris-(2-carboxyethyl)phosphine; 77720; Thermo Fisher Scientific), and subsequently alkylated using 10 mM iodoacetamide. After alkylation, samples were digested with lysyl-endopeptidase (129-02541; Wako Pure Chemical Industries) for 3 h at 37 °C and then further digested with trypsin (4352157; Sigma-Aldrich) for 14 h at 37 °C. Digested peptide samples were analyzed using a nanoscale LC-MS/MS system as previously described[83]. The peptide mixture was

applied to a Mightysil-PR-18 (Kanto Chemical) frit-less column ($45 \times 0.150$ mm ID) and separated using a 0–40% gradient of acetonitrile containing 0.1% formic acid for 80 min at a flow rate of 100 nL/min. Eluted peptides were sprayed directly into a Triple TOF 5600+ mass spectrometer (Sciex). MS/MS spectra were obtained using the information-dependent mode. The MS survey spectrum was acquired in the range of 400 to 1250 $m/z$ for 10 min. For information-dependent acquisition, the 25 highest intensity precursor ions above 50 counts threshold with charge states 2+ and 3+ were selected for MS/MS scans. Each MS/MS experiment set the precursor $m/z$ on a 12 s dynamic exclusion, and the scan range was 100 to 1500 $m/z$ in 100 ms. All MS/MS spectra were searched against protein sequences of NCBI nonredundant human protein dataset (NCBInr RefSeq Release 71, containing 179,460 entries) using the Paragon Method in Protein Pilot software package (Sciex) with following parameters: Cystein Alkylation, iodoacetamide; Digestion, Trypsin; Detected Protein Threshold [Unused ProtScore (Conf.)] >: 5.0 (95.0%)[84]. Protein quantification was performed using the iBAQ method[85] without conversion to absolute amounts using universal proteomics standards (iBQ). The iBQ value was calculated by dividing the sum of the ion intensities of all the identified peptides of each protein by the number of theoretically measurable peptides.

**LC-MS/MS analysis of FLAG-NEK9 immunoprecipitates**. MEFs stably expressing FLAG or FLAG-NEK9 were incubated to confluency, serum-starved for 24 h, and lysed with lysis buffer (50 mM Tris-HCl, pH 7.5, 150 mM NaCl, 1% NP-40, and complete EDTA-free protease inhibitor [03969-21; Nacalai Tesque]). After centrifugation at $17,700 \times g$ for 10 min, the supernatants were incubated with anti-FLAG M2 magnetic beads for 3 h at 4 °C with gentle rotation. The eluted proteins were enzymatically digested according to a phase-transfer surfactant (PTS) protocol[86]. Then, 50-µL eluted samples were mixed with 85 µL of PTS buffer. Samples were reduced with 10 mM dithiothreitol at room temperature for 30 min and alkylated with 50 mM 2-iodoacetamide (804744; Sigma-Aldrich) at room temperature for 30 min. Next, samples were diluted five-fold by adding 50 mM NH$_4$HCO$_3$ solution followed by digestion with 1 µg of Lysyl Endopeptidase (LysC; 121-05063; Wako Pure Chemical Industries) at 37 °C for 4 h. Samples were further digested with 1 µg of trypsin at 37 °C for 8 h. An equal volume of ethyl acetate acidified with 0.5% TFA was added to the digested samples. After centrifugation at $10,000 \times g$ for 10 min twice at room temperature, the aqueous phase containing peptides was collected and dried using a SpeedVac concentrator (Thermo Fisher Scientific). The dried peptides were solubilized in 100 µL of 2% acetonitrile and 0.1% TFA, and the peptide mixture was trapped on a hand-made C18 STAGE tip prepared as previously reported[87]. The trapped peptides were subjected to a previously reported dimethyl-labeling procedure[87]. Subsequently, $^{13}$CH$_2$O and NaBH$_3$CN (light label) were added to the FLAG-only sample. Similarly, $^{13}$CD$_2$O and NaBD$_3$CN (heavy label) were added to the FLAG-NEK9 sample. The dimethyl-labeled peptides left on the tip were eluted with 100 µL of 80% acetonitrile and 0.1% TFA. The light/heavy-labeled eluents were mixed and dried using a SpeedVac concentrator. The sample was dissolved in 2% acetonitrile and 0.1% TFA and loaded onto the LC-MS system with a Q-Exactive MS instrument (Thermo Fisher Scientific) equipped with a nano HPLC system (Advance UHPLC; Bruker Daltonics) and an HTC-PAL autosampler (CTC Analytics) with a trap column ($0.3 \times 5$ mm, L-column, ODS, Chemicals Evaluation and Research Institute). Samples were separated by a gradient using mobile phases A (0.1% formic acid/H$_2$O) and B (0.1% formic acid and 100% acetonitrile) at a flow rate of 300 nL/min (4% to 32% B for 190 min, 32% to 95% B for 1 min, 95% B for 2 min, 95% to 4% B for 1 min, and 4% B for 6 min) with a home-made capillary column (length of 200 mm and inner diameter of 100 µm) packed with 2-µm C18 resin (L-column2, Chemicals Evaluation and Research Institute). Then, the eluted peptides were electrosprayed (1.8–2.3 kV) and introduced into the MS equipment. Data was obtained using the positive ion mode of data-dependent MS/MS (ddMS$^2$) acquisition. Full MS sectra were obtained with a scan range of 350–1800 m/z with 70,000 FWHM resolution at 200 m/z. MS$^2$ spectra were obtained with 17,500 FWHM resolution at 200 m/z. For the ddMS$^2$ acquisition, the 10 highest precursor ions (excluding isotopes of a cluster) above 1.7e4 intensity threshold with charge state from 2+ to 5+ were selected at 4.0 $m/z$ isolation window. A 20 s dynamic exclusion was applied. The obtained raw data were subjected to database search (UniProt, reviewed mouse database as of September 13th, 2018) with Sequest HT algorithm running on Proteome Discoverer 2.2 (Thermo Fisher Scientific). The parameters for database searches were as follows: peptide cleavage was set to trypsin; missed cleavage sites were allowed to be up to two residues; peptide lengths were set to 6–144 amino acids; and mass tolerances were set to 10 ppm for precursor ions and 0.02 Da for fragment ions. For modification conditions, carbamidomethylation at cysteine and dimethylation [H(4)C(2), or H(-2)D(6)$^{13}$C(2)] at lysine and peptide N-terminus were set as fixed modifications. Oxidation at methionine was set as a variable modification. A significance threshold of $p < 0.05$ was applied. Abundances of precursor ions were calculated based on the area of the precursors with Proteome Discoverer 2.2.

**Fluorescence recovery after photobleaching**. MEFs expressing GFP-actin were incubated to confluency and serum-starved for 24 h before analysis. Fluorescence microscopy was performed using a confocal laser microscope (FV3000; Olympus) with a 100 × oil-immersion objective lens (1.40 NA; Olympus) and captured with FluoView software (FV31S-SW, Olympus). The chamber was maintained at 37 °C

and continuously supplied with humidified 5% $CO_2$. Five pre-bleaching images were acquired at 5-s intervals, and circular regions of interest (ROI) with radii of 2 μm were selected and bleached with an 80% power 488-nm laser for 800 ms. Fluorescence recovery was recorded for 250 s by acquiring images at 5 s intervals. Pre- and post-bleach images were taken with 0.5% power. Fluorescence intensity was analyzed by Fiji and its simFRAP plug-in (https://imagej.nih.gov/ij/plugins/sim-frap/index.html). Briefly, the mean intensity values of the ROIs, total image, and background fluorescence were calculated. After background subtraction, the mean intensity values of ROIs were normalized with those of each total image for each time-point.

**Multiple sequence alignment and sequence analysis**. Amino acid sequences of each protein were obtained from the NCBI protein database and aligned using the ClustalW algorithm in MEGA 10.0.5[88]. The disordered regions were predicted using the PSIPRED protein sequence analysis workbench (http://bioinf.cs.ucl.ac.uk/psipred/). The LIR was predicted using iLIR search (https://ilir.warwick.ac.uk/).

**Statistical analysis**. Statistical analysis was performed using GraphPad Prism 8.0 software (GraphPad software). The statistical methods used for each analysis are specified in the figure legends.

**Reporting summary**. Further information on research design is available in the Nature Research Reporting Summary linked to this article.

## Data availability

The datasets generated during and/or analyzed in the current study are available from the corresponding author upon reasonable request. The mass spectrometry proteomics data (LC-MS/MS analysis of GABARAPL1 and GABARAPL1$^{Y49A/L50A}$ immunoprecipitates, and FLAG-NEK9 immunoprecipitates) have been deposited to the ProteomeXchange Consortium via the PRIDE partner repository with the dataset identifiers PXD024290 and PXD024292, respectively. Uniprot database search (reviewed mouse database as of September 13th, 2018) was used for analysis of FLAG-GABARAPL1 and FLAG-GABARAPL1$^{Y49A/L50A}$ immunoprecipitates. NCBI nonredundant human protein dataset (NCBInr RefSeq Release 71, containing 179,460 entries) was used for analysis of FLAG-NEK9 immunoprecipitates. Source data are provided with this paper.

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

## Acknowledgements

We thank Takahide Nagase for his mentorship and encouragement, Hayashi Yamamoto for constructive discussion, Saori Yoshii for mouse sampling, Keiko Igarashi for help with histological examinations, Shoji Yamaoka for pMRX-vector, Teruhito Yasui for pCG-VSV-G and pCG-gag-pol, Robert A. Weinberg for pCMV-VSV-G, Didier Trono for psPAX2, and Keith Mostov for pLKO.1-blast vector. This work was supported by a grant for Exploratory Research for Advanced Technology (grant number JPMJER1702 to N.M.) from the Japan Science and Technology Agency.

## Author contributions

N.M. conceived the project. Y.Y. designed and performed most of the experiments with help from H.C. H.C. performed GABARAPL1-interactome analysis. S.T. generated *Nek9*^W967A/ W967A mice. K.L.O. and H.R.U. performed LC-MS/MS of FLAG-NEK9 immunoprecipitates. Y.Y. and N.M. wrote the manuscript with input from all authors.

## Competing interests

The authors declare no competing interests.
