## [Peer Review File · Nature Communications]

REVIEWER COMMENTS

Reviewer #1 (Remarks to the Author):

In this manuscript, authors identified NIMA-related kinase 9 (NEK9) as a GABARAPs-interacting protein and found that NEK9 and its LC3-interacting region (LIR) are required for primary cilia formation. They further identified and characterized interaction of NEK9 with MYH9, which was implicated in inhibiting ciliogenesis via actin stabilization. Very interestingly, MYH9 accumulates in NEK9 LIR mutant cells and mice, and depletion of MYH9 restored ciliogenesis in NEK9 LIR mutant cells. Therefore, they proposed that NEK9 regulates ciliogenesis by acting as an autophagy adaptor for MYH9.

This study has several interesting findings to provide a novel mechanism of ciliogenesis by NEK9-MYH9 axis in a LIR dependent manner. Overall, this work was well designed and performed at cellular, and biochemical level in vitro and in vivo.

However, there are only minor concerns that authors need to clarify and support author's findings.

1. Authors showed that NEK9 interacts with MYH9 via its C-terminus. How is their interaction regulated during selective autophagy of MYH9 during ciliogenesis? Authors only showed accumulation of MYH9 in LIR mutant cells of NEK9.
2. Is there any possibility that OFD1 associates with MYH9 indirectly? How about the cellular localization pattern of OFD1 and MYH9 in wild-type cells upon starvation?
3. In Fig.5c, immunostaining signal of NEK9 in NEK9 LIR mutant KI cells are very low. However, In Fig.5f, level of NEK9 in NEK9 LIR mutant KI cells is high in Western blot analysis.
4. Authors identified NEK9 as a GABARAPs-interacting protein. In Fig.5c and supplementary Fig.3b, they used LC3 antibody for immunostaining to see colocalization of NEK9, myosinIIA, and LC3. Have authors tried to use GABARAPs antibody instead to LC3 antibody?

Reviewer #2 (Remarks to the Author):

The authors report interesting findings demonstrating that NEK9 is a GABARAP interactor and that its LIR domain is required for primary cilia formation. Altogether the results suggest that NEK9 regulates ciliogenesis by acting as an autophagy receptor for MYH9. The paper is clearly and well written and adds new information on the complex connection between autophagy and cilia.

Maior:

My main concern is that the mutated NEK9 could have an impact on autophagy and this could influence ciliogenesis. The authors showed in suppl Fig 2 that NEK9 does not influence p62 degradation. The authors should study the autophagic flux in NEK9 mutant cells. Is this normal or perturbed? Ideally the authors should count LC3 and WIPI2 puncta and perform WB experiment with and without bafilomycin in Nek9 ko and Nek9W967A MEFs and discuss the results.

In addition, co-localization of NEK9 with FIP200, WIPI2 and LAMP1 should be performed with endogenous antibody against the NEK9 protein

The authors demonstrated that the endogenous NEK9 protein do not form puncta in FIP200 ko cells. To have a more complete story they should analyze also additional cellular model depleted for autophagy genes.

Finally concerning the in vivo studies, the authors should specify the stage at which the Nek9 mutant murine kidneys were studied. Did the authors observe renal cysts at any stage in these mice? Given

the frequency of renal cystic disease in cilia-associated disorders this should be specified. These results should be also discussed in the context of the link between renal cystic disease and ciliogenesis.

Minor:

The interaction between NEK9 and LC3/GABARAP should be moved from suppl fig 1a to the main text

Reviewer #3 (Remarks to the Author):

The authors identify the NIMA-family kinase NEK9 as an interactor of GABARAP proteins, map this interaction and describe that it is required for primary cilia formation in both cultured cells and animals. They relate this to autophagy and NEK9 degradation. Interestingly, animals expressing a mutant form of NEK9 unable to bind to GABARAP show increasing amounts of the kinase, and possibly as a result of this abnormal cilia numbers and structure as well as cell size in the kidney. Although suggested by previous data obtained in NEK9 mutant human fibroblasts, the control of ciliogenesis is a previously unknown function for NEK9. The manuscript furthermore describes an interaction between NEK9 and MYH9/myosin IIA, a negative regulator of ciliogenesis, and suggests that this could mechanistically explain NEK9 importance for ciliogenesis through an action on the actin cytoskeleton.

The data is clear and remarkably extensive (while reading the manuscript I found myself thinking about experiments that the authors in most of the cases immediately describe in the next paragraph or section), as well as well presented and discussed.

Overall the manuscript presents and discusses a number of results that I think will be of interest to the journal readership, and especially to those studying autophagy, cilia and the function of the NIMA family of protein kinases.

I have a few major comments plus some minor ones:

Major:

- It is shown that NEK9 W967A accumulates in MEFs. As this mutant is used extensively in the manuscript, I would suggest to clearly establish that it is indeed not degraded by autophagy, i.e. in response to starvation in MEFs (as in Figure 2F for the wt form of the kinase).
- Supplementary Figure 3L-N should show the expression of the corresponding recombinant protein forms in MEFS and compare it to endogenous NEK9 levels. I would suggest moving this part of Sup. Figure 3 to the main figures (i.e. to Figure 3).
- Would the C-terminal part of NEK9 (i.e. residues ~750-979) be able to complement the lack of NEK9 regarding cilia formation and length? This could be shown in MEFS or HK-2 cells.

Minor:

- line 58: "and lysosomes, and intracellular pathogens." This is probably a typo. Are lysosomes selective autophagy cargos?
- line 69: "Recent evidence suggests that autophagy regulates primary cilia formation bilaterally; cilia regulate autophagy induction, whereas autophagy regulates ciliogenesis ." The sentence is awkward.
- line128: " NEK9 has intrinsically disordered regions (residues 750–891 and 940–979) in the C-terminal region (Fig. 1c).". Indicate how this has been predicted.

- IN MEFS, does bafilomycin A1 result in the accumulation of NEK9 wt to levels similar to those of NEK9 W967A?
- Figure 3D could show some examples of (presumably shorter) cilia in NEK9 W967A cells. A similar comment applies to Fig4.
- line 220: "confirming that NEK9 is degraded by selective autophagy in vivo." Change to "suggesting".
- Supplementary Figure 5 suggests that NEK9 degradation per se is not important for ciliogenesis. Does the western blot correspond to ciliated or exponentially growing cells? or put in another way, do FLAG NEK9 amounts in these cells go down upon cell cycle exit and ciliogenesis?
- NEK7 has been described to be involved in ciliogenesis (Shalom et al. (2008) FEBS Lett 582: 1465–70; Salem et al (2010) Oncogene 29: 4046–57). In principle in non-cycling or interphase cells one would not expect NEK7 (or NEK6) to bind to (then inactive) NEK9 (Regué et al. (2011) J Biol Chem 286: 18118–29), and thus NEK7 would not be degraded together with NEK9. But in Figure 6 it is shown that in serum-starved MEFS NEK7 do in fact interact with NEK9. Wouldn't this result in NEK7 being degraded? The authors could discuss this.
- The causal connection between NEK9, the control of actin dynamics and ciliogenesis is not so clearly established to me. I suggest avoiding sentences such as "we showed that NEK9 functions as a selective autophagy adaptor to degrade MYH9 and promotes ciliogenesis by increasing actin dynamics." (line 377).

Reviewer #4 (Remarks to the Author):

Methodology wise, everything appears to be fine. However, mass spectrometry data, and all associated database search results, must be uploaded to a public repository. Also, the description of the LC-MS/MS data from the GABARAPL dataset is incomplete.

General Responses to Reviewers

We would like to thank the Reviewers for their insightful and constructive comments. We have revised this manuscript according to your suggestions.

The major experimental data that we have added are:

- Intact autophagic flux in *Nek9*^{W967A} MEFs (Supplementary Fig. 3f-h)
- Overtime quantification of NEK9 in *Nek9*^{W967A} MEFs in response to starvation (Supplementary Fig. 3d,e)
- Colocalization between endogenous NEK9 and additional autophagy-related proteins (Supplementary Fig. 1c,f)
- Absence of the interaction and colocalization between OFD1 and MYH9 (Supplementary Fig. 5i and Supplementary Fig. 8a)
- Increased interaction between NEK9 and MYH9 in response to serum starvation (Supplementary Fig. 5k,l)
- Complementation of the defect in ciliogenesis in *Nek9*-KO cells with the C-terminal region of NEK9 (residues 750-979) (Supplementary Fig. 6f-i)

Our point-by-point responses to each Reviewer are attached below. Major changes are indicated in **red** in the manuscript.

Responses to the comments of Reviewer #1

Reviewer #1: In this manuscript, authors identified NIMA-related kinase 9 (NEK9) as a GABARAPs-interacting protein and found that NEK9 and its LC3-interacting region (LIR) are required for primary cilia formation. They further identified and characterized interaction of NEK9 with MYH9, which was implicated in inhibiting ciliogenesis via actin stabilization. Very interestingly, MYH9 accumulates in NEK9 LIR mutant cells and mice, and depletion of MYH9 restored ciliogenesis in NEK9 LIR mutant cells. Therefore, they proposed that NEK9 regulates ciliogenesis by acting as an autophagy adaptor for MYH9. This study has several interesting findings to provide a novel mechanism of ciliogenesis by NEK9-MYH9 axis in a LIR dependent manner.

Overall, this work was well designed and performed at cellular, and biochemical level in vitro and in vivo. However, there are only minor concerns that authors need to clarify and support author's findings.

Response:

We would like to thank this Reviewer for his/her feedback on our manuscript.

1. Authors showed that NEK9 interacts with MYH9 via its C-terminus. How is their interaction regulated during selective autophagy of MYH9 during ciliogenesis? Authors only showed accumulation of MYH9 in LIR mutant cells of NEK9.

Response:

We thank the Reviewer for pointing this out. We measured how the interaction between NEK9 and MYH9 changes in response to serum starvation by immunoprecipitation analysis. The result suggests that their interaction was strengthened in serum starvation conditions (Supplementary Fig. 5k,l). This should serve the purpose because ciliogenesis is induced in response to serum starvation. (Lines 297-299)

Supplementary Fig. 5k,l

Supplementary Fig. 5 NEK9 is a selective autophagy adaptor for MYH9.

k, Immunoprecipitation using MEFs stably expressing FLAG or FLAG-NEK9 after serum starvation (6 h). **l**, Quantification of the intensity ratio of MYH9 bands (IP / Input) in **k**. Data represent the mean \pm SEM of three independent experiments.

2. Is there any possibility that OFD1 associates with MYH9 indirectly? How about the cellular localization pattern of OFD1 and MYH9 in wild-type cells upon starvation?

Response:

We thank the Reviewer for pointing this out. In the previous manuscript, we showed that FLAG-OFD1 does not form a complex with MYH9 (Supplementary Fig. 8a (now 8b)). We have confirmed it using FLAG-MYH9-expressing cells (new Supplementary Fig. 5i). As recommended, we also performed colocalization immunofluorescence analysis and found no colocalization between MYH9 and OFD1 (new Supplementary Fig. 8a). Consistently, a recent paper revealed that OFD1 has a LIR motif and interacts directly with ATG8 proteins¹. We think MYH9 and OFD1 associate with autophagic membranes independently. (Lines 369 and 410-413)

Supplementary Fig. 5i

Supplementary Fig. 5 NEK9 is a selective autophagy adaptor for MYH9.

i, Immunoprecipitation using MEFs stably expressing FLAG or FLAG-MYH9.

Supplementary Fig. 8a

Supplementary Fig. 8 Autophagic degradation of NEK9–MYH9 and OFD1 is required for primary cilia formation.

a, Immunofluorescence microscopy of wild-type MEFs expressing GFP-MYH9, showing that MYH9 does not colocalize with OFD1. Scale bars, 10 μ m and 3 μ m (insets).

3. In Fig.5c, immunostaining signal of NEK9 in NEK9 LIR mutant KI cells are very low. However, In Fig.5f, level of NEK9 in NEK9 LIR mutant KI cells is high in Western blot analysis.

Response:

We apologize that the contrast adjustment of NEK9 fluorescence was not properly standardized between normal cells and NEK9 cells in Fig. 5c. We have now adjusted the contrast of the immunofluorescence microscopy data in new Fig. 5c.

Figure. 5c

Fig. 5 NEK9 is a selective autophagy adaptor for MYH9. c,

Immunofluorescence microscopy of wild-type and *Nek9*^{W967A} MEFs stably expressing GFP-MYH9 after serum-starvation (4 h). Cells were stained with anti-NEK9 and anti-LC3 antibodies. Scale bars, 10 μ m and 3 μ m (insets).

4. Authors identified NEK9 as a GABARAPs-interacting protein. In Fig.5c and supplementary Fig.3b, they used LC3 antibody for immunostaining to see colocalization of NEK9, myosinIIA, and LC3. Have authors tried to use GABARAPs antibody instead to LC3 antibody?

Response:

We thank the Reviewer for pointing this out. As recommended, we performed immunofluorescence analysis using GABARAP antibody and found clear colocalization with endogenous NEK9 (Supplementary Fig. 1c) and GFP-MYH9 (Supplementary Fig. 5j). (Lines 151-152, 293)

Supplementary Fig. 1c

Supplementary Fig. 1 NEK9 associates with autophagic membranes. c, Immunofluorescence microscopy of endogenous NEK9 and GABARAP in wild-type MEFs after starvation (2 h).

Supplementary Fig. 5j

Supplementary Fig. 5 NEK9 is a selective autophagy adaptor for MYH9. j, Immunofluorescence microscopy of MEFs expressing GFP-MYH9 after serum starvation (2 h). Cells were stained with anti-GABARAP antibody.

Responses to the comments of Reviewer #2

The authors report interesting findings demonstrating that NEK9 is a GABARAP interactor and that its LIR domain is required for primary cilia formation. Altogether the results suggest that NEK9 regulates ciliogenesis by acting as an autophagy receptor for MYH9. The paper is clearly and well written and adds new information on the complex connection between autophagy and cilia.

Maior:

My main concern is that the mutated NEK9 could have an impact on autophagy and this could influence ciliogenesis. The authors showed in suppl Fig 2 that NEK9 does not influence p62 degradation. The authors should study the autophagic flux in NEK9 mutant cells. Is this normal or perturbed? Ideally the authors should count LC3 and WIPI2 puncta and perform WB experiment with and without bafilomycin in Nek9 ko and Nek9W967A MEFs and discuss the results.

Response:

We thank the Reviewer for valuable suggestions. In this revised manuscript, we have examined autophagy flux in NEK9 KO and mutant (*Nek9*^{W967A}) cells using an autophagic flux reporter (GFP-LC3-RFP) and by the LC3 conversion assay, which are generally more quantitative and objective than counting LC3 and WIPI2 puncta. These results show that general autophagy flux is not affected in *Nek9-KO* and *Nek9*^{W967A} cells (Supplementary Fig. 3f–h). (Lines 195-200)

Supplementary Fig. 3f-h

Supplementary Fig. 3 Selective autophagy of NEK9 is required for cilia formation.

f, Quantitative autophagic flux assays. Wild-type, *Nek9*-KO, and *Nek9*^{W967A} (clone #7) MEFs stably expressing the GFP-LC3-RFP reporter were stimulated by Torin 1, an inhibitor of mTOR. Autophagic activity was quantified as the ratio of the GFP to RFP fluorescence. Data were collected from 2,000 cells for each cell-type. Solid bars indicate the medians, boxes the interquartile range (25th to 75th percentile), and whiskers the 10th to 90th percentile.

g, Immunoblotting of wild-type, *Nek9*-KO, and *Nek9*^{W967A} (clone #7) MEFs under nutrient-rich conditions or after starvation (2 h) with or without 100 nM bafilomycin A₁ (baf A₁). **h**, Quantification of the intensity of the NEK9 bands. Data represent the mean \pm SEM of three independent experiments.

In addition, co-localization of NEK9 with FIP200, WIPI2 and LAMP1 should be

performed with endogenous antibody against the NEK9 protein

Response:

We thank the Reviewer for this suggestion. We performed immunofluorescence microscopy to evaluate colocalization of endogenous NEK9 with FIP200, WIPI2, and LAMP1 (Supplementary Fig. 1f). Our results show endogenous NEK9 colocalizes with these proteins (please note that the endogenous NEK9 antibody gives many noisy background signals). (Lines 151-153, 162)

Supplementary Fig. 1f

Supplementary Fig. 1 NEK9 associates with autophagic membranes.

f, Immunofluorescence microscopy of endogenous NEK9 and FIP200, WIPI2, LAMP1 in wild-type MEFs after starvation (2 h).

The authors demonstrated that the endogenous NEK9 protein do not form puncta in FIP200 ko cells. To have a more complete story they should analyze also additional cellular model depleted for autophagy genes.

Response:

We thank the Reviewer for this constructive comment. We performed immunofluorescence microscopy in *Atg3*-KO MEFs and confirmed that NEK9 does not form punctate structures even in starvation conditions (Supplementary Fig. 1g,h). (Line 155)

Supplementary Fig. 1g,h

Supplementary Fig. 1 NEK9 associates with autophagic membranes.

g, Immunofluorescence microscopy of wild-type and *Atg3*-KO MEFs expressing GFP-NEK9 and mRuby3-GABARAP after starvation (2 h). **h**, Quantification of the number of GFP-NEK9 puncta in **g**. Data were collected from 100 cells for each condition. Solid bars indicate the medians, boxes the interquartile range (25th to 75th percentile), and whiskers the 10th to 90th percentile. *p*-value corresponds to two-tailed Mann–Whitney test. Scale bars, 10 μ m and 3 μ m (insets).

Finally concerning the in vivo studies, the authors should specify the stage at which the Nek9 mutant murine kidneys were studied. Did the authors observe renal cysts at any stage in these mice? Given the frequency of renal cystic disease in cilia-associated disorders this should be specified. These results should be also discussed in the context of the link between renal cystic disease and ciliogenesis.

Response:

We thank the Reviewer for this constructive suggestion. We used five-month-old wild-type (WT) and *Nek9*^{W967A/W967A} mice and three-month-old *Atg5*^{+/+} and *Atg5*^{-/-}; *NSE-Atg5* mice. In the revised manuscript, we have added the age information in the legends to Fig. 4 and Supplementary Fig. 4 and 5. We could not find any renal cysts in *Nek9*^{W967A/W967A} mice up to twelve-month-old (data not shown). Consistently, renal cysts have never been reported in autophagy-deficient mice²⁻⁴.

Although impaired degradation of NEK9–MYH9 by selective autophagy suppresses ciliogenesis and induces secondary cellular hypertrophy in renal tubular cells, cilia managed to grow in about half of these cells (Fig. 4b; Supplementary Fig. 4b), which may prevent cyst formation. We have included these points in the Discussion part of this revised manuscript. (Lines 442-448)

Minor:

The interaction between NEK9 and LC3/GABARAP should be moved from suppl fig 1a to the main text

We thank the Reviewer for this suggestion. We have moved the immunoprecipitation data to new Fig. 1d.

Responses to the comments of Reviewer #3

The authors identify the NIMA-family kinase NEK9 as an interactor of GABARAP proteins, map this interaction and describe that it is required for primary cilia formation in both cultured cells and animals. They relate this to autophagy and NEK9 degradation. Interestingly, animals expressing a mutant form of NEK9 unable to bind to GABARAP show increasing amounts of the kinase, and possibly as a result of this abnormal cilia numbers and structure as well as cell size in the kidney.

Although suggested by previous data obtained in NEK9 mutant human fibroblasts, the control of ciliogenesis is a previously unknown function for NEK9. The manuscript furthermore describes an interaction between NEK9 and MYH9/myosin IIA, a negative regulator of ciliogenesis, and suggests that this could mechanistically explain NEK9 importance for ciliogenesis through an action on the actin cytoskeleton.

The data is clear and remarkably extensive (while reading the manuscript I found myself thinking about experiments that the authors in most of the cases immediately describe in the next paragraph or section), as well as well presented and discussed.

Overall the manuscript presents and discusses a number of results that I think will be of interest to the journal readership, and especially to those studying autophagy, cilia and the function of the NIMA family of protein kinases.

Response:

We thank the Reviewer for these supportive comments.

I have a few major comments plus some minor ones:

Major:

- It is shown that NEK9 W967A accumulates in MEFs. As this mutant is used extensively in the manuscript, I would suggest to clearly establish that it is indeed not degraded by autophagy, i.e. in response to starvation in MEFs (as in Figure 2F for the wt form of the kinase).

Response:

We thank the Reviewer for this suggestion. We evaluated the amount of NEK9 in response to starvation in *Nek9*^{W967A} MEFs. The result suggests that NEK9^{W967A} is not degraded by autophagy (Supplementary Fig. 3d,e). (Lines 192-193)

Supplementary Fig. 3d,e

Supplementary Fig. 3 Selective autophagy of NEK9 is required for cilia formation. **d**, Wild-type and *Nek9*^{W967A} (clone #7) MEFs were incubated under starvation conditions with or without 100 nM bafilomycin A₁ for the indicated time. Whole-cell lysates were subjected to immunoblotting. **e**, Quantification of the intensity of the NEK9 bands in **d**. Data represent the mean \pm SEM values of three independent experiments.

- Supplementary Figure 3L-N should show the expression of the corresponding recombinant protein forms in MEFs and compare it to endogenous NEK9 levels. I would suggest moving this part of Sup. Figure 3 to the main figures (i.e. to Figure 3).

Response:

We thank the Reviewer for pointing this out. We have added western blot data to show the expression level of the endogenous NEK9 and the overexpressed NEK9 recombinants (Supplementary Fig. 3q). As recommended, we have moved previous Supplementary Figure 3L-N to the main figures (Fig. 3j-l).

Supplementary Fig. 3q

Supplementary Fig. 3 Selective autophagy of NEK9 is required for cilia formation. **q**, Immunoblotting of wild-type and *Nek9*-KO MEFs expressing indicated constructs, corresponding to Fig. 3j. Asterisk indicates endogenous NEK9 band.

- Would the C-terminal part of NEK9 (i.e. residues ~750-979) be able to complement the lack of NEK9 regarding cilia formation and length? This could be shown in MEFS or HK-2 cells.

Response:

We thank the Reviewer for asking this important question. We determined whether the C-terminal region of NEK9 including the LIR and residues 973–979 is sufficient to facilitate ciliogenesis. The expression of GFP-tagged NEK9 750–979 restored the defects in ciliogenesis in *Nek9*-KO MEFs (Supplementary Fig. 6f-i). This result suggests that the C-terminal region of NEK9 including the LIR and residues 973–979 is sufficient to regulate ciliogenesis. (Lines 332-337)

Supplementary Fig. 6f-i

Supplementary Fig. 6 NEK9-mediated selective autophagy of MYH9 is required for primary cilia formation. **f**, Immunoblotting of *Nek9*-KO MEFs expressing indicated constructs. Data are representative of three independent experiments. **g**, Immunofluorescence microscopy of *Nek9*-KO MEFs expressing indicated constructs after serum starvation (24 h). **h**, The frequency of ciliated cells in **g**. Data represent the mean \pm SEM of five independent experiments (300 cells were counted in each experiment). **i**, Quantification of cilia length in **g**. Data were collected from 100 ciliated cells for each cell-type. Solid bars indicate the medians, boxes the interquartile range (25th to 75th percentile), and whiskers the 10th to 90th percentile. *p*-values correspond to Tukey's multiple comparisons tests in **h**, **i**; **p* < 0.0001. Scale bars, 10 μ m and 3 μ m (insets).

Minor:

- line 58: "and lysosomes, and intracellular pathogens." This is probably a typo. Are lysosomes selective autophagy cargos?

Response:

We thank the Reviewer for pointing this out. With all due respect to the Reviewer, it is well established that damaged lysosomes can be selectively eliminated by autophagy, which is termed lysophagy⁵. We believe that it is not inappropriate to say that lysosomes are selective autophagy cargos. To help readers, we have added this specific reference (Ref #17 in the manuscript).

- line 69: "Recent evidence suggests that autophagy regulates primary cilia formation bilaterally; cilia regulate autophagy induction, whereas autophagy regulates ciliogenesis." The sentence is awkward.

Response:

We apologize for the odd sentence we have presented. We have changed the sentence as follows;

"Recent evidence suggests that the relationship between autophagy and ciliogenesis is bidirectional; cilia regulate autophagy induction, whereas autophagy regulates ciliogenesis." (Lines 69-71)

- line128: "NEK9 has intrinsically disordered regions (residues 750–891 and 940–979) in the C-terminal region (Fig. 1c)". Indicate how this has been predicted.

Response:

We thank the Reviewer for pointing this out. As we described in the Methods section, we used the PSIPRED protein sequence analysis workbench to predict disordered regions. We have added this information in the main text of this revised manuscript (Lines 129-131).

- IN MEFS, does bafilomycin A1 result in the accumulation of NEK9 wt to levels similar to those of NEK9 W967A?

Response:

We thank the Reviewer for asking this point. We compared the amount of NEK9 between bafilomycin A₁-treated wild-type cells and *Nek9*^{W967A} cells (Supplementary Fig. 3d,e). Even after 9-hour treatment of bafilomycin A₁, NEK9 did not accumulate in wild-type cells as much as in *Nek9*^{W967A} cells. One possible explanation is that, compared to the half-life of NEK9 protein, the 9-

hour treatment was too short to cause NEK9 accumulation. Since longer bafilomycin A₁ treatment demonstrates substantial cell-toxicity, we could not observe its prolonged effect. (Lines 192-193)

Supplementary Fig. 3d,e

Supplementary Fig. 3 Selective autophagy of NEK9 is required for cilia formation. **d**, Wild-type and *Nek9^{W967A}* (clone #7) MEFs were incubated under starvation conditions with or without 100 nM bafilomycin A₁ for the indicated time. Whole-cell lysates were subjected to immunoblotting. **e**, Quantification of the intensity of the NEK9 bands in **d**. Data represent the mean \pm SEM values of three independent experiments.

- Figure 3D could show some examples of (presumably shorter) cilia in NEK9 W967A cells. A similar comment applies to Fig4.

Response:

We thank the Reviewer for this suggestion. We have changed the image to a more appropriate one containing shorter cilia (Fig. 3d, Fig. 4b).

Fig. 3d

Fig. 3 Selective autophagy of NEK9 is required for primary cilia formation.
d, Immunofluorescence microscopy of wild-type or *Nek9*^{W967A} MEFs after serum starvation (24 h). Centrosomes and primary cilia were stained with anti-pericentrin and anti-ARL13B antibodies, respectively. Scale bars, 10 μm and 3 μm (insets).

Fig. 4b

Fig. 4 Selective autophagy of NEK9 is required for primary cilia formation in mouse kidneys. b, Immunohistochemistry of the cortical region of kidneys

from five-month-old wild-type and *Nek9*^{W967A/W967A} mice using LTL-FITC (the lumen of proximal-tubular cells) and anti-ARL13B antibody (primary cilia). Scale bars, 40 μm and 5 μm (insets).

- line 220: “confirming that NEK9 is degraded by selective autophagy in vivo.”
Change to “suggesting”.

Response:

We thank the Reviewer for this comment and have changed the sentence accordingly. (Line 230)

- Supplementary Figure 5 suggests that NEK9 degradation per se is not important for ciliogenesis. Does the western blot correspond to ciliated or exponentially growing cells? or put in another way, do FLAG NEK9 amounts in these cells go down upon cell cycle exit and ciliogenesis?

Response:

We thank the Reviewer for asking this question. The western blot shown in Supplementary Fig. 5a corresponds to exponentially growing cells. We think that the level of overexpressed NEK9 was too high to detect a clear reduction in response to starvation. We believe that overexpressed FLAG-NEK9 is also degraded under cilia-inducing serum starvation conditions.

- NEK7 has been described to be involved in ciliogenesis (Shalom et al. (2008) FEBS Lett 582: 1465–70; Salem et al (2010) Oncogene 29: 4046–57). In principle in non-cycling or interphase cells one would not expect NEK7 (or NEK6) to bind to (then inactive) NEK9 (Regué et al. (2011) J Biol Chem 286: 18118–29), and thus NEK7 would not be degraded together with NEK9. But in Figure 6 it is shown that in serum-starved MEFS NEK7 do in fact interact with NEK9. Wouldn't this result in NEK7 being degraded? The authors could discuss this.

Response:

We apologize for not having explained the rationale for using NEK7 in Fig. 6. As mentioned by this Reviewer, NEK7 interacts with NEK9 predominantly during mitosis, and therefore is not a suitable positive control for IP experiments performed during interphase. As we did not have another appropriate positive control that interacts with NEK9 during interphase, we used NEK7. We performed this experiment (Fig. 6b) after 4 h serum starvation, when the cell

cycle might not be completely arrested. We think that NEK9–NEK7 interaction was detected in cells that were still in the mitotic phase. As NEK7 did not accumulate in *Nek9*^{W967A/W967A} mice (Fig. 1 for Reviewers), it is unlikely that NEK7 is degraded together with NEK9.

Fig. 1 for Reviewers. Immunoblotting of the indicated organs of five-month-old wild-type (WT) and *Nek9*^{W967A/W967A} mice (KI). Data are representative of three biologically independent replicates.

- The causal connection between NEK9, the control of actin dynamics and ciliogenesis is not so clearly established to me. I suggest avoiding sentences such as “we showed that NEK9 functions as a selective autophagy adaptor to degrade MYH9 and promotes ciliogenesis by increasing actin dynamics.” (line 377).

We thank the Reviewer for pointing this out. We have changed the sentence as follows; "we showed that NEK9 functions as a selective autophagy adaptor to degrade MYH9. Considering the previous report that MYH9 suppresses actin dynamics prior to ciliogenesis, our data suggested that selective autophagy of NEK9–MYH9 promotes ciliogenesis by increasing actin dynamics." (Lines 396-400)

Reviewer #4 (Remarks to the Author): Methodology wise, everything appears to be fine. However, mass spectrometry data, and all associated database search results, must be uploaded to a public repository. Also, the description of the LC-MS/MS data from the GABARAPL dataset is incomplete.

Response:

We thank the Reviewer for valuable and constructive comments. As recommended, we have deposited the mass spectrometry proteomics data (LC-MS/MS analysis of GABARAPL1 and GABARAPL1^{Y49A/L50A} immunoprecipitates, and FLAG-NEK9 immunoprecipitates) to the ProteomeXchange Consortium via the PRIDE⁶ partner repository with the dataset identifiers PXD024290 and PXD024292, respectively. Also, we expanded the description of LC-MS/MS in the Method section as follows;

Liquid chromatography-tandem mass spectrometry (LC-MS/MS) analysis of GABARAPL1 and GABARAPL1^{Y49A/L50A} immunoprecipitates

FLAG-GABARAPL1 and FLAG-GABARAPL1^{Y49A/L50A} plasmids were prepared as described above. FLAG-GABARAPL1 and FLAG-GABARAPL1^{Y49A/L50A} interacting proteins were immunoprecipitated from HEK293T cells and identified by LC-MS/MS analysis using a high-performance LabDroid system at Robotic Biology Institute (<https://rbi.co.jp/en/>). Briefly, HEK293T cells transiently expressing FLAG-GABARAPL1 and FLAG-GABARAPL1^{Y49A/L50A} were lysed with lysis buffer (20 mM HEPES-NaOH, pH 7.5, containing 1% digitonin, 150 mM NaCl, 50 mM NaF, 1 mM Na₃VO₄, 5 mg/mL leupeptin, 5 mg/mL aprotinin, 3 mg/mL pepstatin A, and 1 mM phenylmethylsulfonylfluoride) and centrifuged at 20,000 × *g* for 10 min to remove insoluble materials. The supernatants were subjected to immunoprecipitation using anti-FLAG M2 magnetic beads (M8823; Sigma-Aldrich). Precipitated immunocomplexes were washed three times in washing buffer (10 mM HEPES-NaOH, pH 7.5, 150 mM NaCl, and 0.1% Triton X-100) and eluted with FLAG peptides. The samples obtained were subjected to trichloroacetic acid precipitation. The resulting pellets were dissolved in 0.1 M ammonium bicarbonate (pH 8.8) containing 7 M guanidine hydrochloride, reduced using 5 mM TCEP (tris-(2-carboxyethyl)phosphine; 77720; Thermo Fisher Scientific), and subsequently alkylated using 10 mM iodoacetamide. After alkylation, samples were digested with lysyl-endopeptidase (129-02541; Wako Pure Chemical Industries) for 3 h at 37°C and then further digested with

trypsin (4352157; Sigma-Aldrich) for 14 h at 37°C. Digested peptide samples were analyzed using a nanoscale LC-MS/MS system as previously described⁷. The peptide mixture was applied to a Mightysil-PR-18 (Kanto Chemical) frit-less column (45 x 0.150 mm ID) and separated using a 0–40% gradient of acetonitrile containing 0.1% formic acid for 80 min at a flow rate of 100 nL/min. Eluted peptides were sprayed directly into a Triple TOF 5600+ mass spectrometer (Sciex). MS/MS spectra were obtained using the information-dependent mode. Up to 25 precursor ions above an intensity threshold of 50 counts/s were selected for MS/MS analyses from each survey scan. All MS/MS spectra were searched against protein sequences of NCBI nonredundant human protein dataset (NCBI RefSeq Release 71, containing 179,460 entries) using the Protein Pilot software package (Sciex). Protein quantification was performed using the iBAQ method without conversion to absolute amounts using universal proteomics standards (iBQ). The iBQ value was calculated by dividing the sum of the ion intensities of all the identified peptides of each protein by the number of theoretically measurable peptides.

References

1. Morleo, M. *et al.* Regulation of autophagosome biogenesis by OFD1-mediated selective autophagy. *EMBO J* **40**, e105120 (2021).
2. Wang, S., Livingston, M.J., Su, Y. & Dong, Z. Reciprocal regulation of cilia and autophagy via the MTOR and proteasome pathways. *Autophagy* **11**, 607–616 (2015).
3. Yoshii, S.R. *et al.* Systemic Analysis of Atg5-Null Mice Rescued from Neonatal Lethality by Transgenic ATG5 Expression in Neurons. *Dev Cell* **39**, 116–130 (2016).
4. Kimura, T. *et al.* Autophagy protects the proximal tubule from degeneration and acute ischemic injury. *J Am Soc Nephrol* **22**, 902–913 (2011).
5. Papadopoulos, C., Kravic, B. & Meyer, H. Repair or Lysophagy: Dealing with Damaged Lysosomes. *J Mol Biol* **432**, 231–239 (2020).
6. Perez-Riverol, Y. *et al.* The PRIDE database and related tools and resources in 2019: improving support for quantification data. *Nucleic Acids Res* **47**, D442–D450 (2019).
7. Natsume, T. *et al.* A direct nanoflow liquid chromatography–tandem mass spectrometry system for interaction proteomics. *Anal Chem* **74**, 4725–4733 (2002).

REVIEWERS' COMMENTS

Reviewer #1 (Remarks to the Author):

The authors have largely addressed the detailed questions I raised in my previous review. I think this is a commendable advancement and therefore, I recommend its publication.

Reviewer #2 (Remarks to the Author):

I congratulate the authors for the revisions. All my comments were satisfactorily addressed and I recommend the paper for publication

Brunella Franco

Reviewer #3 (Remarks to the Author):

The authors have addressed all my concerns, adding a substantial amount of data to the manuscript. In my opinion now the authors' claims are adequately supported and the manuscript can thus be accepted for publication without further revisions. Congratulations on an excellent work.

Reviewer #4 (Remarks to the Author):

The authors addressed one major concern by uploading their data to a public repository and adding a data availability statement to the manuscript. However, the methods are still not complete. For instance, the database search parameters are not given for the GABARAPL1 IP experiments (no parent or fragment ion tolerances, variable modifications, or fixed modifications). Mass spec acquisition parameters for this same experiment are incomplete as well, as are the mass spec acquisition parameters for the data acquired on the Q-Exactive. I recommend fixing these omissions before publication.

REVIEWERS' COMMENTS

Response

Reviewer #1 (Remarks to the Author):

The authors have largely addressed the detailed questions I raised in my previous review. I think this is a commendable advancement and therefore, I recommend its publication.

Response:

We thank this Reviewer for this warm comment

Reviewer #2 (Remarks to the Author):

I congratulate the authors for the revisions. All my comments were satisfactorily addressed and I recommend the paper for publication

Response:

We thank this Reviewer for this kind comment

Reviewer #3 (Remarks to the Author):

The authors have addressed all my concerns, adding a substantial amount of data to the manuscript. In my opinion now the authors' claims are adequately supported and the manuscript can thus be accepted for publication without further revisions. Congratulations on an excellent work.

We thank this Reviewer for this kind and supportive comment

Reviewer #4 (Remarks to the Author):

The authors addressed one major concern by uploading their data to a public repository and adding a data availability statement to the manuscript. However, the methods are still not complete. For instance, the database search parameters are not given for the GABARAPL1 IP experiments (no parent or fragment ion tolerances, variable modifications, or fixed modifications). Mass spec acquisition

parameters for this same experiment are incomplete as well, as are the mass spec acquisition parameters for the data acquired on the Q-Exactive. I recommend fixing these omissions before publication.

Response:

We thank this Reviewer for this constructive comment. In accordance with this suggestion, we have expanded the description of MS analysis in the method, including the database search parameters and mass spec acquisition parameters.